# Optogenetically stimulating intact rat corticospinal tract post-stroke restores motor control through regionalized functional circuit formation

A.S. Wahl[1,2,3], U. Büchler[4], A. Brändli[1,2], B. Brattoli[4], S. Musall[1], H. Kasper[1], B.V. Ineichen[1,2], F. Helmchen[1], B. Ommer[4] & M.E. Schwab[1,2]

Current neuromodulatory strategies to enhance motor recovery after stroke often target large brain areas non-specifically and without sufficient understanding of their interaction with internal repair mechanisms. Here we developed a novel therapeutic approach by specifically activating corticospinal circuitry using optogenetics after large strokes in rats. Similar to a neuronal growth-promoting immunotherapy, optogenetic stimulation together with intense, scheduled rehabilitation leads to the restoration of lost movement patterns rather than induced compensatory actions, as revealed by a computer vision-based automatic behavior analysis. Optogenetically activated corticospinal neurons promote axonal sprouting from the intact to the denervated cervical hemi-cord. Conversely, optogenetically silencing subsets of corticospinal neurons in recovered animals, results in mistargeting of the restored grasping function, thus identifying the reestablishment of specific and anatomically localized cortical microcircuits. These results provide a conceptual framework to improve established clinical techniques such as transcranial magnetic or transcranial direct current stimulation in stroke patients.

[1] Brain Research Institute, University of Zurich, Winterthurerstr. 190, 8057 Zurich, Switzerland. [2] Department of Health Sciences and Technology, ETH Zurich, Winterthurerstr. 190, 8057 Zurich, Switzerland. [3] Central Institute of Mental Health, University of Heidelberg, J5, 68159 Mannheim, Germany. [4] Computer Vision Group, Interdisciplinary Center for Scientific Computing (IWR), University of Heidelberg, Mathematikon (INF 205), 69120 Heidelberg, Germany. A.S. Wahl and U. Büchler contributed equally to this work. B. Ommer and M.E. Schwab jointly supervised this work. Correspondence and requests for materials should be addressed to A.S.W. (email: wahl@hifo.uzh.ch) or to M.E.S. (email: schwab@hifo.uzh.ch)

Destruction of large cortical areas, as typically caused in cortical strokes, induces plastic remodeling and reorganization of neuronal connections and functions throughout the central nervous system (CNS)[1–3]. Where and how new connections grow, which areas and circuitries are either activated or repressed and what they contribute in terms of functional recovery is poorly known. While patients or animals with small strokes often show high levels of spontaneous recovery associated with rewiring of perilesional areas, large stroke lesions with > 60% of unilateral cortex destruction have a poor prognosis with very limited capacity for regaining lost functions[4, 5]. Remodeling processes are often associated with sprouting of spared axons, which innervate denervated target areas and form new circuits for the recovery of lost functions[5–7]. For large strokes, specific experimental manipulations were shown to promote internal repair mechanisms such as pharmacological treatments, rehabilitative training strategies, stem cell transplantation, or brain stimulation[8–11]. However, the success and feasibility of these approaches were often variable due to unspecific activation or inhibition processes of undefined cell types producing undesired side effects[12, 13].

In order to design optimized therapeutic strategies, it is crucial to understand the interaction between external, therapeutic manipulation with the internal repair and reorganization processes on the level of distinct CNS circuits. Furthermore, unbiased detailed, quantitative behavioral readouts for comparing the healthy with the stroke-impaired and the rehabilitation-induced condition are necessary to assess the level of recovery (restoration of function vs. forms of compensation) and to identify the most successful treatment. This analysis should be automatic and non-invasive to avoid interference with the recovery process.

In clinical trials and experimental studies of large ischemic strokes, the contralesional pre- and sensorimotor cortex as well as the intact contralesional corticospinal tract (CST) have been identified as the brain structures involved most closely in reorganizational processes underlying potential recovery of impaired motor functions[14, 15]. While the interpretation of enhanced activity levels in the contralesional motor cortex (M1) remains controversial[1, 16–19], sprouting of the intact contralesional CST terminating in the stroke denervated hemi-spinal cord (either ipsilaterally projecting fibers or midline crossing fibers), has been described after various therapeutic and rehabilitative interventions[5, 9, 20–23]. Positive correlations between the amounts of newly sprouting CST fibers and the level of motor recovery as well as the direct demonstration that pharmacogenetic silencing of these sprouted fibers abolished the recovered skilled reaching performance[22, 24] suggests a beneficial role of corticospinal rewiring.

Here we tested a new optogenetic activation protocol as a novel potential therapeutic approach after stroke. As we used a large sensorimotor stroke, our study focused on the manipulation and circuit investigation of the intact, contralesional hemisphere, as the main region where plastic remodeling and reorganizational processes are likely to be found which mediate recovery of movements in the paretic side. We selectively stimulated intact CST neurons using optogenetics in the contralesional motor cortex after a large sensorimotor stroke. A novel, unsupervised computer vision-based analysis helped to evaluate fine-scale modulation of rat forepaw grasping behavior before the stroke and during the rehabilitation and recovery process under the different rehabilitative interventions. We find that optogenetic CST activation, combined with subsequent motor training, leads to full recovery of forelimb function through CST axon remodeling in the spinal cord, similar to a previously established anti-Nogo-A immunotherapy with subsequent training[22]. We furthermore investigated whether axonal rewiring re-establishes specific reorganization patterns to fully compensate for lost functions. In animals with full motor recovery after the combined immunotherapeutic/training protocol, we selectively inactivated distinct subsets of newly sprouted fibers of intact CST neurons using inhibitory optogenetics during the forelimb grasping task. Indeed, using automatic visual analysis of paw posture and kinematics, we found specific functional deficits induced by optogenetic silencing of premotor and M1 subregions of the contralesional cortex. Our findings highlight the great potential of specific neural activation protocols in combination with motor training for the recovery of skilled motor functions after stroke.

## Results

**Optogenetically stimulating the intact corticospinal tract**. We aimed at testing the therapeutic effect of optogenetically activating the intact corticospinal tract of the contralesional hemisphere after a large stroke in adult rats. After training the intact, prelesioned animals in the single pellet grasping task for skilled forelimb function[25] and selectively expressing Channelrhodopsin-2 (ChR2) in motor cortex neurons projecting to the spinal cord via a dual viral approach (Fig. 1b, Methods), rats received a photothrombotic stroke destroying the premotor and the sensorimotor cortex, corresponding to the preferred paw (Methods, Supplementary Fig. 1). Rats were then randomized into four different rehabilitation groups (Fig. 1a): In two of the groups, the intact corticospinal tract on the contralesional side was optogenetically stimulated 3 times/day for 2 weeks starting at day 3 after the stroke. Light delivery occurred through three optic fibers implanted to cover the pre- and primary contralesional motor cortex (Supplementary Fig. 2). The stimulation paradigm consisted of $3 \times 1$ min stimulation at 10 Hz of 473 nm wavelength LED light with 3-min intervals in between[8]. We observed visible movements, partially with rhythmic jerking, of the corresponding limbs and freezing behavior, especially during the first 7 days after stroke (Supplementary Movie 1)—an effect that vanished during the second week after stroke. Whereas in the "OptoStim group" stimulation of the corticospinal path was the only treatment, animals in the "OptoStim/Training group" were additionally trained on the impaired paw in the single pellet grasping task with 100 reaches per day starting after the 2 weeks of light stimulation up to 4 weeks after stroke (Fig. 1a). We here used the sequential approach of first optogentic stimulation of the intact corticospinal tract followed by intensive training based on our previous experience that early combination of two plasticity stimulating approaches could be detrimental[22]. In the "Delayed Training" group, animals just received the single pellet grasping training between 2 and 4 weeks after stroke. The fourth "Spontaneous recovery" group of rats did not receive any special treatment but was tested at the same time points as the other groups for skilled grasping function.

Optogenetically stimulating the intact corticospinal tract resulted in greatly improved forelimb function in both optical stimulation groups (Fig. 1c, "OptoStim group" $52.9 \pm 7.3\%$, "OptoStim/Training group" $53.0 \pm 8.9\%$, $p < 0.05$, two-way repeated measures ANOVA with post hoc Bonferroni, Supplementary Fig. 5) already 7 days after insult compared to the "Delayed Training" or the untrained, unstimulated "Spontaneous recovery" group. Importantly, animals with sequential optical stimulation and training (OptoStim/Training) reached success rates 5 weeks after stroke comparable to healthy baseline conditions, while animals with intensive, delayed training showed significantly lower levels of regained forelimb function up to the end of the rehabilitative paradigm (Fig. 1c "OptoStim/Training" group $93.3 \pm 15.8\%$, "Delayed Training" group $34.0 \pm 13.7\%$, $p < 0.001$, two-way repeated measures ANOVA with post hoc Bonferroni).

We exposed all animals at the end of the rehabilitative schedule (5–6 weeks post stroke) to novel tasks of skilled forelimb usage to assess non-task-specific recovery of motor function. Animals in both stimulation groups—with or without additional training—performed significantly better than the other two groups in narrow beam crossing (Fig. 1d, "OptoStim" group $54.7 \pm 4.0\%$, "OptoStim/Training" group $58.8 \pm 4.7\%$ vs. $37.3 \pm 2.6\%$ in the "Spontaneous recovery" and $35.2 \pm 2.7\%$ in the "Delayed Training" group, $p < 0.05$, two-way repeated measures ANOVA with post hoc Bonferroni) and in horizontal ladder crossing (Fig. 1e, "OptoStim" group $54.7 \pm 2.8\%$, "OptoStim/Training" group $57.0 \pm 2.4\%$ vs. $38.6 \pm 2.8\%$ in the "Spontaneous recovery" and $35.4 \pm 3.4\%$ in the "Delayed Training" group, $p < 0.05$, two-way repeated measures ANOVA with post hoc Bonferroni).

These results indicate that optogenetic activation promoted non-task-specific recovery of forelimb function. Histological analysis of mCherry-positive cells (Fig. 1f) showed about 39% of layer 5 neurons in M1 to be labeled, indicating that a large proportion of the CST neurons expressed ChR2 and were presumably activated by the light stimulation.

**Full recovery vs. compensation of lost functions.** Recently, we showed that combining 2 weeks of the growth-promoting anti-Nogo-A immunotherapy followed by 2 weeks of intensive training after a large sensorimotor cortex stroke resulted in full recovery of lost forelimb function[22]. We repeated this protocol and directly compared the efficiency of this combinational

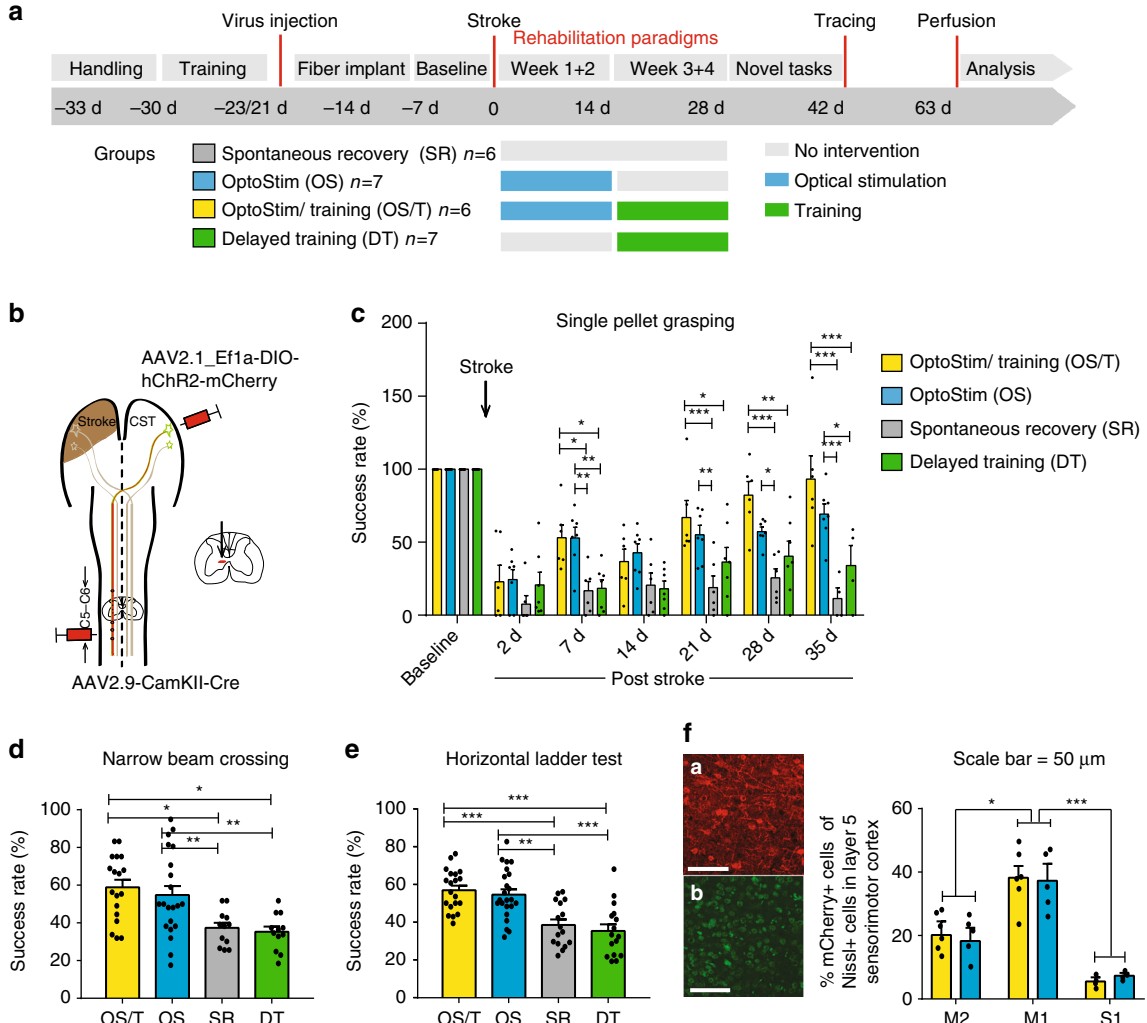

**Fig. 1** Specific optogenetic stimulation of the intact corticospinal tract originating from the contralesional hemisphere results in robust recovery of lost forelimb function after stroke. **a** Timeline for the four different treatment groups. **b** Schematic diagram of viral vector injections into the contralesional motor cortex (AAV2.1-Ef1a-DIO-hChR2(t159C)-mCherry) and the corresponding cervical hemi-spinal cord (AAV2.9-CamKII0.4.Cre.SV40) to specifically express Channelrhodopsin-2 (ChR2) in the intact corticospinal neurons. **c** Success rates in the single pellet grasping task relative to baseline (100%; intact, trained) 2 days to 5 weeks after a large, unilateral photothrombotic stroke to the sensorimotor cortex of the preferred paw. The "OptoStim/Training" and "OptoStim" groups showed significant improvement of skilled forelimb function compared to the other groups. Both stimulation groups ("OptoStim" and "OptoStim/Training" group) also performed significantly better in novel tasks to assess recovery of forelimb function such as the narrow beam task (**d**) and the horizontal ladder crossing task (**e**) tested at 5 weeks after stroke. Shown are average success rates of three consecutive trials. **f** Representative images of mCherry-positive cells (**a**) in the sensorimotor cortex in comparison to Nissl-positive cells (**b**) (**a**, **b**: scale bars = 50 μm) as well as quantification of mCherry-positive cells in percentage of Nissl-positive cells in layer 5 of pre- (M2), primary motor cortex (M1), and primary sensory cortex (S1) for both "stimulation" groups. Data are presented as means ± s.e.m.; statistical evaluation was carried out with two-way (for **c**) and one-way (for **d**, **e**) ANOVA repeated measure followed by Bonferroni post hoc, and with paired $t$-test, two-tailed for F; asterisks indicate significances: *$p < 0.05$, **$p < 0.01$, and ***$p < 0.001$

therapy to that of optogenetic stimulation of the intact CST with subsequent training (OptoStim/Training group). A cohort of rats receiving the immunotherapy during weeks 1 and 2 after the stroke followed by grasping training (weeks 3 and 4 after stroke,

Fig. 2a, "Anti-Nogo/Training" group) was tested during the course of recovery in comparison to animals of the "OptoStim/ Training" group and "Spontaneous recovery group" (from Fig. 1). "Anti-Nogo/Training" and "OptoStim/Training" animals revealed

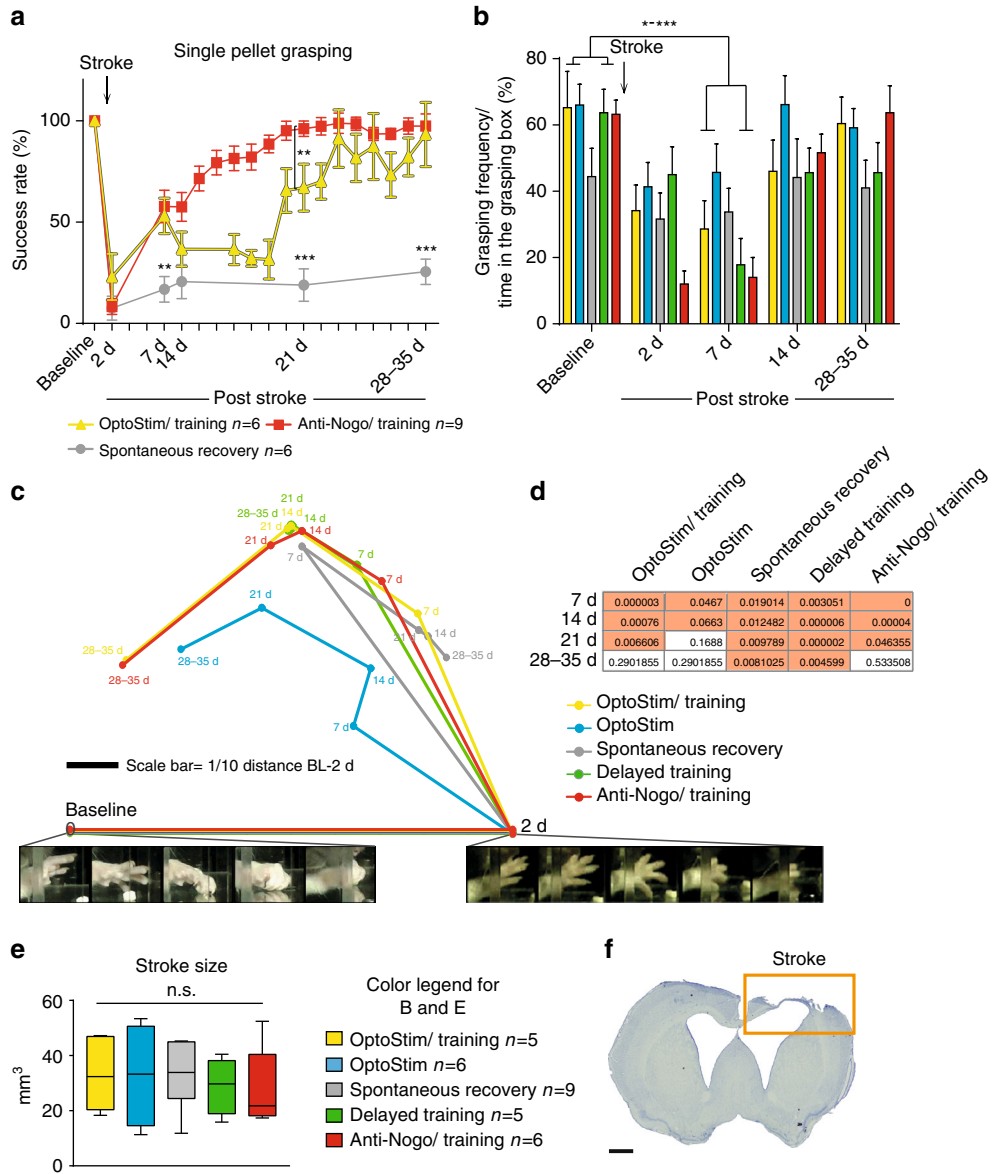

**Fig. 2** Brief daily optogenetic stimulation of the intact corticospinal tract leads to full recovery of forelimb function similar to a growth-promoting therapy combined with training. **a** Stroke animals that received anti-Nogo-A immunotherapy (Anti-Nogo/Training) or optogenetic CST stimulation (OptoStim/ Training group) followed by rehabilitative training performed significantly better in the single pellet grasping task than stroke animals without treatment (Spontaneous recovery group, same group as in Fig. 1) even 1 week after stroke. **b** Computer vision-based automatic analysis of overall grasping activity of animals from all five treatment groups (from Figs. 1 and 2a) during their time in the grasping box showed that all animals were less likely to perform single pellet grasping within the first week after stroke, but the frequency of grasping activity was back close to baseline levels in all groups 4–6 weeks after stroke. **c** Automatic video-based grasping analysis. Paws were tracked and represented by comparing them against a codebook of prototypical postures. Thereafter, the similarity of grasping sequences to baseline and 2d grasps was computed using a sequence matching approach and averaged for each recording session of each cohort. The resulting two distances (capturing the relative degradation compared to baseline and the improvement compared to 2d) yield a relative location that is then visualized by triangulation. During the recovery phase, the grasping behavior of the "Stimulation/Training" and "Anti-Nogo/Training" groups becomes progressively more similar to the baseline configuration. In contrast, the delayed training and spontaneous cohorts show no substantial improvement. **d** Statistical significance analysis of **c**. Grasps at 7d, 14d, 21d and 28–35 are compared to the baseline. A K–S test then reveals that the "Spontaneous recovery" and "Delayed Training" cohorts show indeed no significant recovery ($p > 0.05$) at the rehabilitative end point (28–35d post stroke), as opposed to the three other treatment cohorts, whose behavior shows no significant difference compared to baseline anymore. **e** No significant difference was found for the stroke lesion size among all five treatment groups examined. **f** Representative coronal section showing the lesioned motor cortex in Nissl staining (8 weeks after stroke, scale bar = 1 mm). Data are presented as means ± s.e.m.; statistical evaluation was carried out with two-way ANOVA (**a**, **b**) and one-way ANOVA (**e**) repeated measure followed by Bonferroni post hoc; asterisks indicate significances: *$p < 0.05$, **$p < 0.01$, and ***$p < 0.001$

a significant regain of forelimb function already at 1 week after stroke (Fig. 2a). However, full recovery of prelesion levels was reached in the fourth week after stroke for the "OptoStim/Training" group, while animals in the "Anti-NogoTraining" cohort achieved success scores similar to baseline levels already 3 weeks after stroke. Very much in contrast, animals without treatment (Spontaneous recovery group) plateaued at a success rate below 30% throughout the post-stroke phase (Fig. 2a success rate at 4 weeks: 97.4 ± 6.0% "Anti-Nogo/Training" group,

93.3 ± 15.8% "OptoStim/Training" group, 25.5 ± 6.2% "Spontaneous recovery" group; $p < 0.001$, two-way repeated measures ANOVA with post hoc Bonferroni). Stroke volume analysis of all rehabilitative groups revealed no difference in lesion size between the treatment groups (Fig. 2e, one-way ANOVA repeated measure followed by Bonferroni post hoc).

We next investigated which of the five treatment paradigms promoted true restoration or only a mere compensation of forelimb function. In case of restoration, complete recovery of the

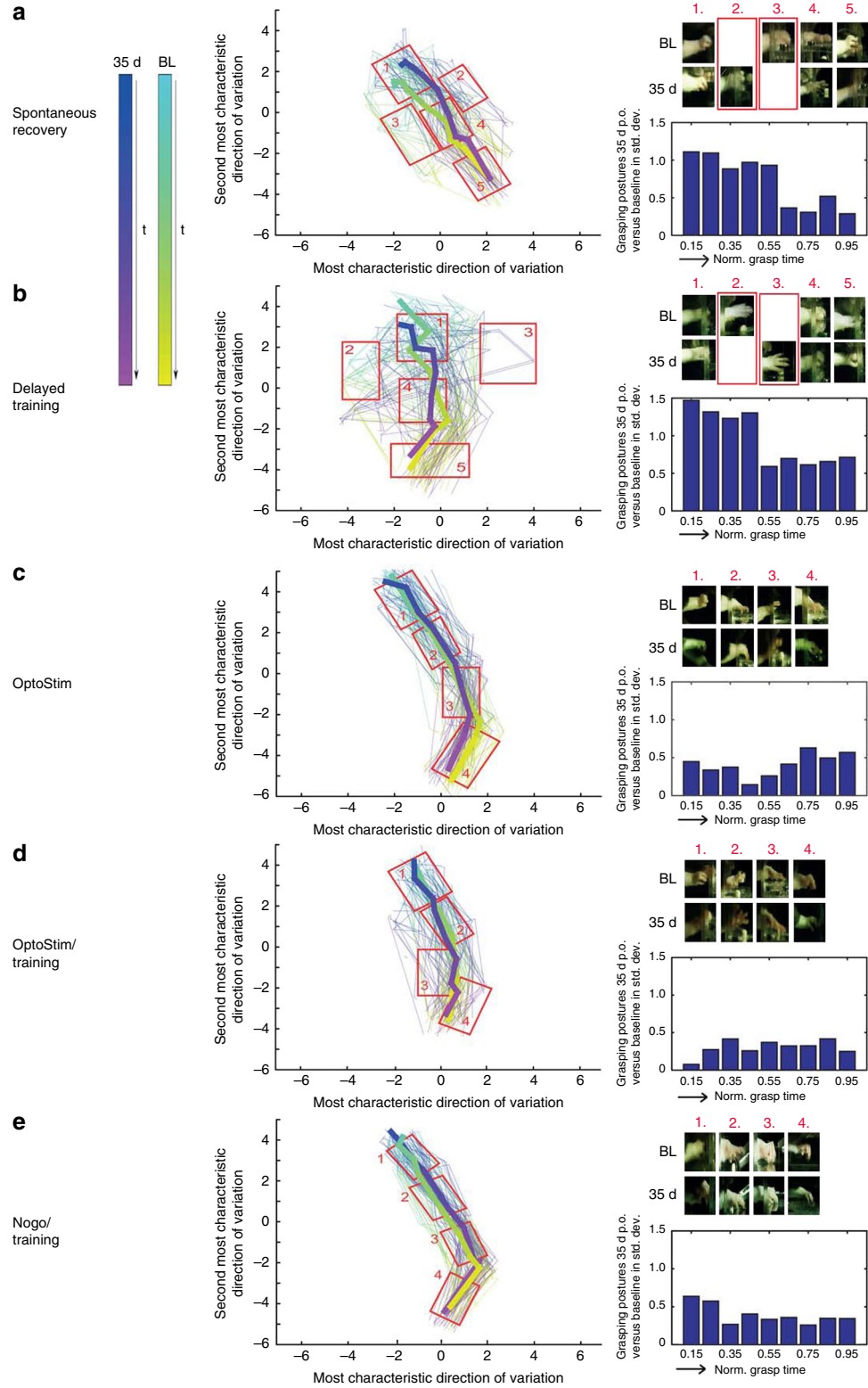

pre-stroke grasping kinematics and the sequence of postural motifs was achieved, whereas compensation lead to the substitution of the original movement sequence by other movements. We utilized high-resolution video recordings of single pellet grasping before stroke and during 4–5 weeks of rehabilitation. Grasp actions were automatically evaluated with a novel computer vision algorithm for unsupervised behavior analysis (see "Methods" section for details). Not only does this algorithm extract the spatial trajectories of a grasp but also analyses its kinematics, i.e., it compares the sequences of individual postures that comprise each grasp. After an initial paw detection and tracking, the paw posture at each time point was represented by comparing it with a dictionary of prototypical poses extracted from successful grasping moves. Afterwards, grasping sequences were compared using a sequence matching approach, yielding a distance between grasps based on the similarity of their respective sequences of postures. To determine the sensitivity of our approach, we first aimed at detecting differences of grasping quality during the course of learning the single pellet grasping task (over 24 days before "baseline" recordings prior to stroke). Our algorithm was able to classify even small differences in the healthy condition during the course of skilled motor learning (see Supplementary Fig. 3). Using this method, we then assessed how frequently animals showed grasping activity as opposed to being idle (Fig. 2b): Grasping frequency of all animals was reduced within the first week after stroke. Reduced grasping frequency can be due to decreased motivation or reduced grasping velocity because of proximal shoulder and elbow impairments. While grasping frequency quickly recovered to baseline levels within 2 weeks for all cohorts, the skilled behavior did not as investigated in Fig. 2c. In this figure, we precisely compared grasps at different stages after stroke against two references, the skilled baseline behavior as well as impaired activity 2 days after stroke (one example from either set is presented below the figure). For each cohort and time point, we thus obtained an average distance of all grasps of that session to the baseline and 2 days grasping. Using these two distances, grasping in each recording session was localized by triangulation in a two-dimensional plot: Sessions appearing toward the left are more similar to baseline, those at the right are closer to the state right after stroke. Those at the top are further away from both baseline and stroke, meaning that there was compensation, which altered the behavior from what it was right after stroke, but also increased the difference to baseline. We see that the "OptoStim/Training," "Anti-Nogo/Training," and "OptoStim" groups exhibit reorganization of the behavior. Until 3 weeks after stroke, their grasping is significantly altered with respect to 2 days and baseline. Thereafter, at 28–35 days grasping of these three treatment groups again becomes significantly more

similar to what it was at baseline. In Fig. 2d, a K–S test was conducted to confirm if the discrepancy of posture sequences between grasping at a particular time after stroke and those at baseline was significant ($p < 0.05$). Four weeks after stroke, the "OptoStim/Training," "Anti-Nogo/Training," and "OptoStim" groups showed no significant dissimilarity to baseline anymore, implying a full restoration of forelimb function for these groups. In contrast, the other two cohorts (Spontaneous recovery and Delayed Training) exhibited forms of compensation, since their behavior changed significantly after stroke without becoming similar to baseline.

Figure 3 puts into relation individual grasping sequences at baseline and 35 days after stroke within each cohort to discover which phases of a grasp differed most. As in the previous experiment, paws were tracked during each grasp and their individual postures were represented by comparing them against a dictionary of reference postures. These distance vectors were then projected in a two-dimensional graph using a low-dimensional distance preserving embedding[26]. Each grasp was displayed as a polygonal chain beginning at cyan (baseline) and blue (35 days). Additionally, averaging all postures of the same phase of a grasp yielded a mean grasping contour (broad curves). For each cohort, the plots showed which postures were unique to baseline or 35 days. For spontaneous recovery and delayed training, these differences occurred predominantly at mid-grasp around paw closure (cutout 2, 3). The detailed posture analysis over time revealed too short grasps for the "Spontaneous recovery" group (Fig. 3a) 28–35d after stroke while animals in the "Delayed Training" group failed to supinate the paw for the targeting of the pellet even at the same end point (Fig. 3b). For the other three cohorts, no such differences were visible. Moreover, the bar charts in Fig. 3 showed for different phases of a grasp (beginning at 0 and ending at 1) the average distance of posture between baseline and 35 days relative to the standard deviation of the baseline. Overall, the "OptoStim" (C), "OptoStim/Training" (D), and "Anti-Nogo/Training" (E) groups revealed an average deviation of grasping posture between 35d and baseline that was 2–3 times less compared to the "Spontaneous recovery" and "Delayed Training" groups. We also included a sham-operated control group in our analysis (Supplementary Fig. 4) and did not find any posture differences between baseline and 35 days after sham surgery.

## Optogenetic stimulation promotes corticospinal fiber growth.
The intact corticospinal tract normally innervates the spinal cord half opposite to the one which has lost its cortical input due to the stroke. Only a few fibers cross the spinal cord midline.

**Fig. 3** Analysis of grasping trajectories indicating the efficiency of each rehabilitative schedule to promote recovery of impaired grasping function after stroke. For all five rehabilitative paradigms **a**–**e** grasping kinematics (left panel) at baseline (cyan-yellow trajectories) and 35d after stroke (blue-magenta trajectories) were compared. The grasping kinematics are represented by comparing the individual postures of a grasping sequence against a dictionary of prototypical postures and visualizing the resulting distance vectors using a two-dimensional distance preserving embedding. The two axes correspond to major variations in posture represented in units of baseline standard deviation (SD). Postures of an individual grasp are linked using a polygonal chain (100 grasping events for baseline and 100 grasping events for 35d after stroke per group) and color indicates the progress of a grasp over time from beginning to end. The superimposed lines represent the average reprogression of the individual grasps for baseline and 35d post insult. Midpoint is when the paw is closest to the pellet. Moreover, we sampled postures from different areas of this embedding to show differences/similarities between 35d after stroke and baseline for the different cohorts. The right bar panel summarizes how much behavior at baseline and at 35d after stroke differs at different stages of grasping. Grasps are split into five disjoint time intervals and we compute the distance between baseline and 35d relative to the SD of the baseline. On top right of each group **a**–**e** selected frames of the grasping trajectory are depicted at the same time points (1.–4./5.) for baseline (BL) and 35 days post stroke highlighting the divergence/convergence at the end point of the rehabilitative treatment compared to baseline. Animals in the "Delayed Training" group **b** failed to regain the supination motion for the final targeting of the pellet 35d after stroke compared to baseline (>1 SD for the first two disjoint time intervals, right panel bars), while animals in the "Spontaneous recovery" group **a** still showed too short grasps (>1 SD for the first time interval, right panel bars) even 35d after stroke (as indicated in the red boxes for 2. and 3. for the two cohorts)

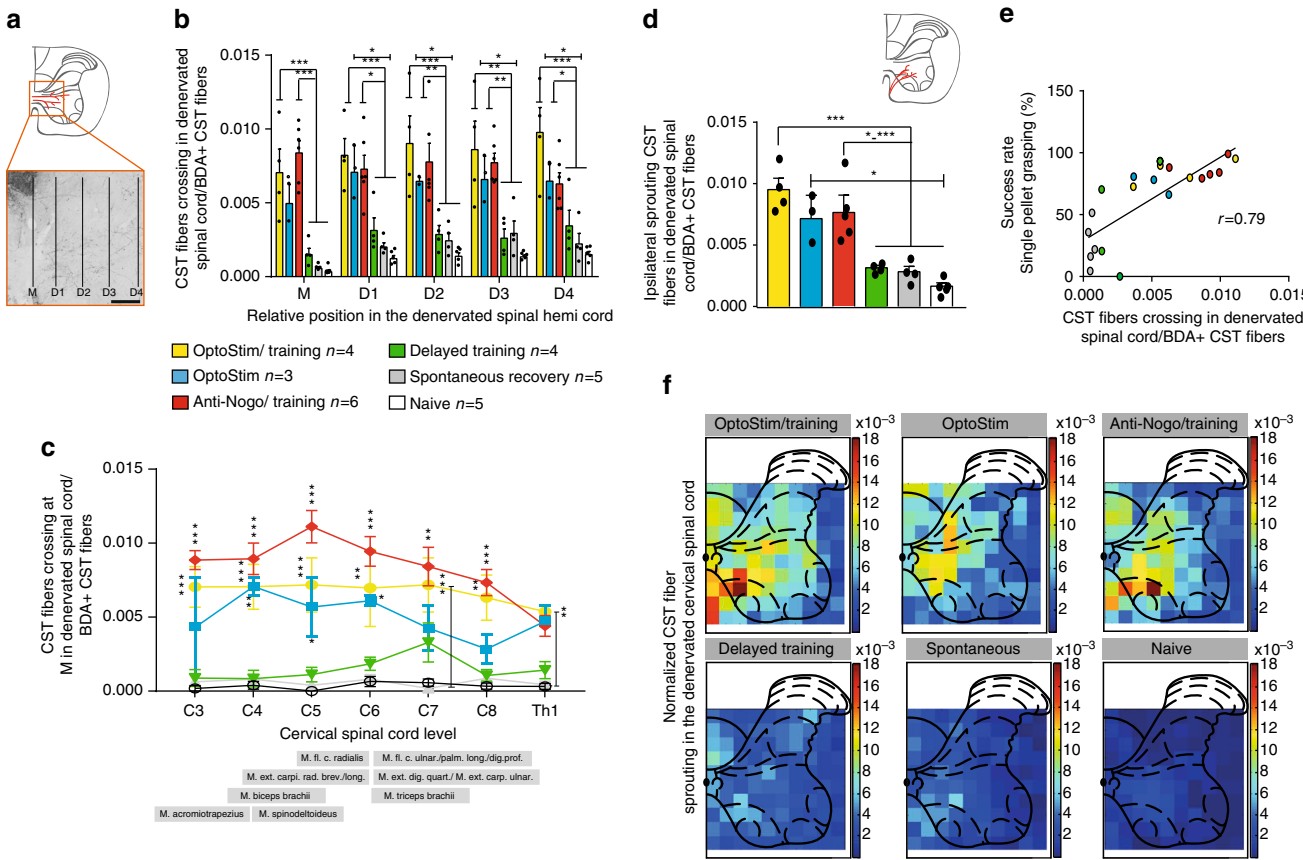

**Fig. 4** Corticospinal fiber sprouting from intact to denervated spinal hemi-cord is enhanced by early cortical stimulation. **a** Corticospinal tract fibers from the intact, contralesional motor cortex cross the spinal cord midline (M) and grow into the corticospinally denervated gray matter for various distances (D1–4, scale bar = 200 μm). **b** Values were significantly higher in the stimulation groups and the anti-Nogo-A group. **c** Segment-specific analysis of midline crossing fibers showed that the two stimulation groups and the "Anti-Nogo/Training" group had significantly more midline crossing fibers in the more rostral cervical spinal cord where motoneuron pools controlling proximal forelimb muscles are located, whereas fiber sprouting in the more caudal cervical spinal cord where motoneuron pools for the distal muscles are located was less pronounced compared to the "Spontaneous recovery" and "Delayed training" groups as well as in naive animals. **d** Sprouting of pre-existing ipsilateral CST fibers in the gray matter of the cervical enlargement after stroke was also most pronounced in the "Anti-Nogo/Training" group as well as in the "OptoStim/Training" group. **e** There was a positive correlation between the amount of newly sprouting CST fibers and the level of recovered grasping function of the impaired forelimb 4–5 weeks after stroke (r = 0.79, Spearman correlation, p < 0.0001). **f** Average false color-coded heat maps of CST fiber sprouting densities in the denervated spinal cord at C4 level. In both stimulation groups and the "Anti-Nogo/Training" group, CST fibers in particular sprouted in lamina 6/7 of the ventral horn. Asterisks represent statistical significance. Data are presented as means ± s.e.m.; statistical evaluation was carried out with two-way ANOVA repeated measure followed by Bonferroni post hoc, asterisks indicate significances: *p < 0.05, **p < 0.01, and ***p < 0.001

We investigated if our paradigm of brief, repeated daily optogenetic stimulation of the contralesional, intact corticospinal tract was sufficient to induce corticospinal fiber sprouting across the midline to innervate the denervated spinal cord, a phenomenon which has been described in rats treated with growth-promoting anti-Nogo antibodies[9, 22]. Corticofugal fibers from the contralesional motor cortex were labeled by anterograde transport of BDA 5–6 weeks after stroke (Fig. 4a). We found the greatest number of midline crossing fibers in the "OptoStim/Training" and "OptoStim," as well as the "Anti-Nogo/Training" group (Fig. 4b and Supplementary Fig. 5). The number of fibers grown into the stroke denervated spinal hemi-cord in these three groups was significantly higher than in other groups when counted at the midline of the cervical spinal cord (Fig. 4b, "M"), but also within the gray matter of the motor laminae 6 and 7 (Fig. 4b, D1–4). A segment-specific analysis along the cervical spinal cord of midline crossing fibers at (M) revealed enhanced corticospinal rewiring throughout the denervated spinal cord for the "OptoStim/ Training" and "Anti-Nogo/Training" groups at C3–C8 vs. the other groups (Fig. 4c; p < 0.01, two-way repeated measures

ANOVA with post hoc Bonferroni). Midline crossing was slightly less pronounced in the "Stimulation" alone group (Fig. 4c). Apart from midline crossing corticospinal fibers, CST fibers can also reach the denervated spinal cord uncrossed, as sprouting, pre-existing ipsilaterally projecting axons. We counted the number of intersections of collaterals of these ipsilaterally projecting fibers at the gray/white matter boundary of the ventral funiculus as a measure of gray matter innervation by the ventral CST. We again found that both stimulation groups and the "Anti-Nogo/ Training" group revealed the highest amount of fiber growth in the ventral CST (Fig. 4d). There was also a positive correlation between the amount of midline crossing CST fibers and the level of skilled motor recovery of the impaired paw 4–5 weeks after stroke (Fig. 4e).

These data show that three short daily periods of brief optogenetic stimulation of the intact corticospinal tract applied over 2 weeks (plus training) are sufficient to induce robust axonal growth into the denervated hemi-cord. Density of sprouting BDA-labeled CST fibers in the denervated spinal cord was up to 16–18 times higher, in particular in lamina 6/7 of the ventral

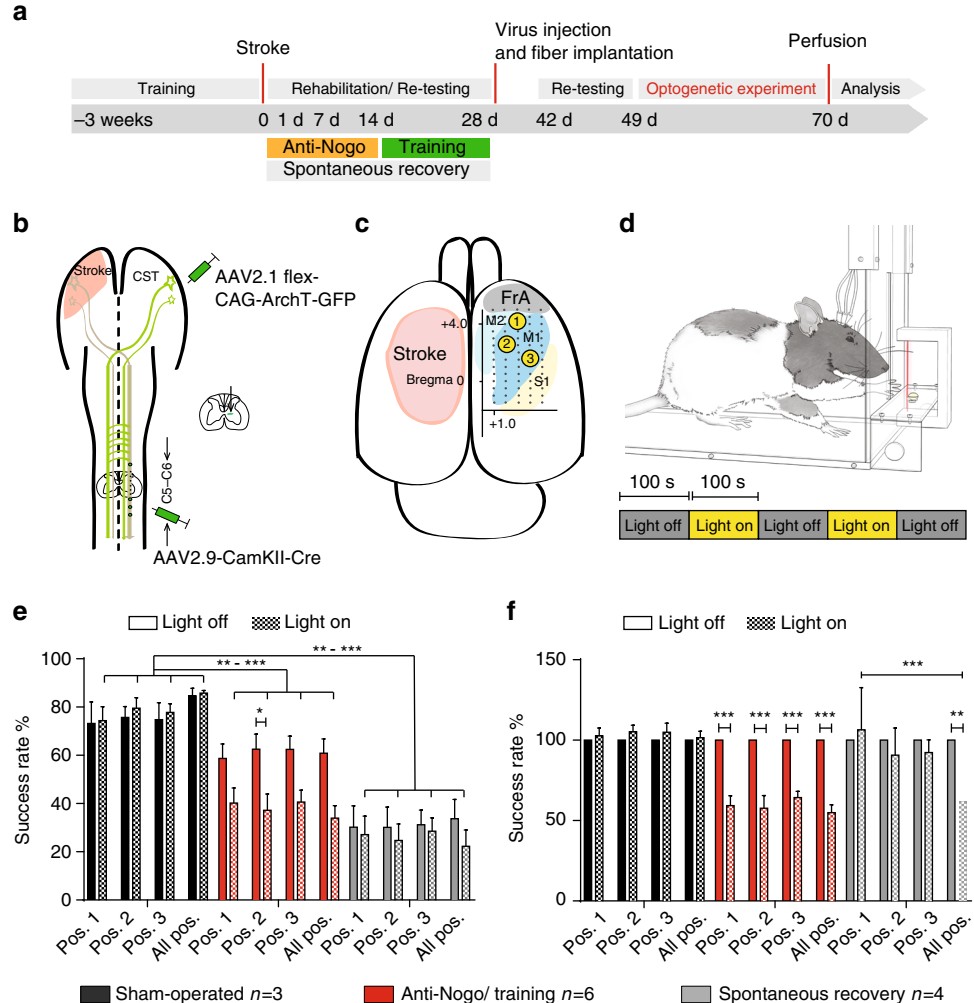

**Fig. 5** Optogenetic silencing of subsets of contralesional corticospinal neurons results in decline of recovered grasping function in animals with excellent recovery. **a** Experimental time schedule. **b** Scheme of vector injections into the contralesional motor cortex (AAV2.1-Flex-ArchT-GFP) and the stroke denervated cervical spinal cord (AAV2.9-CamKII0.4.Cre.SV40). **c** Illustration of the location of the three optical fiber implants: implant #1 is localized in between pre- (M2) and primary motor cortex (M1), implant #2 covers the center of the main forelimb motor cortex (M1), implant #3 lies close to the primary sensory cortex (S1). (FrA frontal area). **d** Experimental setup for optogenetic inactivation during single pellet grasping: Optical fibers are connected to the three optical implants covering parts of the pre- and sensorimotor cortex. When the rats start to grasp for the first sugar pellet, a light barrier is activated which, after a lag time of 100 sec, starts the laser for another 100 s. **e** Optogenetic silencing of rerouted, ipsilaterally projecting corticospinal neurons from the center of the contralesional M1 forelimb motor cortex (Pos. 2/implant #2) results in a significant decline of the recovered grasping function in animals of the "Anti-Nogo/Training" group, while light activation did not perturb grasping performance in control, "sham-operated" animals. **f** Normalizing grasping success rates under "light-on" conditions to success rates for "light-off" conditions for each individual animal per group confirms these results and shows that optogenetic silencing through all three optical fibers also negatively influences grasping success for the "Spontaneous recovery" group. Data are presented as means ± s.e.m.; statistical evaluation was carried out with two-way ANOVA repeated measure followed by Bonferroni post hoc for **d** and **e**, asterisks indicate significances: $*p < 0.05$, $**p < 0.01$, and $***p < 0.001$

horn, in the "OptoStim/Training" and "Anti-Nogo/Training" group compared to BDA-positive fibers in intact (Naive) animals (Fig. 4f). In addition, animals in the "OptoStim" group revealed a 9–13× higher BDA-positive CST fibers density, e.g., in lamina 4/5 compared to intact animals.

**Loss of regained grasping features by cortical silencing.** The importance of the intact CST and in particular of its midline crossing fibers at spinal cord level for the recovery of lost motor functions after a large cortical stroke has been demonstrated in several recent studies[21–23, 27]. Growth pattern and targeting of these midline crossing fibers mainly to lamina 6/7 of the denervated spinal hemi-cord seems crucial for motor recovery[22]. However, the precise origin of these fibers in the premotor, rostral motor field (M2), the main motor forelimb area (M1) or the

sensory forelimb area (S1) remained unclear. In order to address this question, we selectively inactivated subsets of midline crossing corticospinal fibers originating from these three cortical regions during the single pellet grasping task to test their function for specific kinematic aspects of the regained grasping action. We used animals with excellent functional recovery that had received 2 weeks of anti-Nogo immunotherapy followed by 2 weeks of intensive training (Anti-Nogo/Training group) and compared them to animals with a low level of recovery without treatment (Spontaneous recovery group) (Fig. 5a). Animals of the "OptoStim/Training" group were not used here due to technical reasons (vector injection protocols) and possible interference of co-activating and silencing optogenetic vectors. Four weeks after stroke, we injected a Cre-expressing adeno-associated virus (AAV2.9) into segments C5 and C6 of the stroke denervated

cervical hemi-spinal cord[28], followed by injections of the Cre-dependent AAV2.1 vector carrying the inhibitory light-driven outward proton pump Arch T into the contralesional pre- and sensorimotor cortex (Fig. 5b). We then chronically implanted three optical fibers over the cortical surface in such a way that optical fiber #1 targeted parts of the pre- and rostral motor cortex M2, implant #2 was stereotactically positioned above the principal forelimb motor cortex (M1), and implant #3 covered also parts of the primary sensory cortex (Fig. 5c). We also implanted optical fibers into "Sham-operated" naive animals without virus injections to confirm that light stimulation alone did not influence the grasping behavior. For the optogenetic silencing of distinct subsets of corticospinal fibers during the grasping task animals were put in the grasping box and connected via long optical fibers to three lasers (532 nm weavelength), enabling optical silencing at each implant location independently (Fig. 5d). The first grasp for a sugar pellet activated a light barrier, which initiated the three lasers after a time lock of 100 s and kept them on with continuous laser light for 100 s. After a lag time of 100 s with laser lights off, the next grasp initiated another phase of laser light activation. We either stimulated only one position (position 1 vs. position 2 vs. position 3, Fig. 5c) or all three positions at the same time during a grasping session. We found that optical silencing midline crossing CST fibers originating from the contralesional forelimb motor cortex resulted in a drop of performance for all cortical areas silenced in animals of the "Anti-Nogo/Training" group. However, this decrease reached

significance only for M1 (Fig. 5e, "Anti-Nogo/Training" group success rate at position #2 $62.5 \pm 6.2\%$ during "light-off" vs. $37.2 \pm 6.9\%$ success rate during "light-on," $p < 0.05$, two-way repeated measures ANOVA with post hoc Bonferroni). The low level of grasping performance in the "Spontaneous recovery" group did not change significantly by the regional cortical inactivation (Fig. 5e). The grasping success of the "Sham-operated" animals remained unaltered upon light stimulation and animals did not show signs of distress during light stimulation or by the presence of the optic fibers (Fig. 5e). Normalizing grasping success rates of the "light-on" relative to the "light-off" condition individually per animal and per cortical position of stimulation revealed for the animals with excellent recovery ("Anti-Nogo/Training" group) a significant decline of grasping function upon stimulation at each individual position as well as for all positions together (Fig. 5f, relative loss of function in premotor area (position #1) 40.7%; in M1 (position #2) 42.3%; in lateral M1-S1 (position #3) 35.6%; all positions together: 45.0%, $p < 0.001$, two-way repeated measures ANOVA with post hoc Bonferroni). In contrast, only inhibition at all three implant positions at the same time caused a decrease of grasping success rates in animals with already poor recovery of forelimb function ("Spontaneous recovery" group, relative loss of function upon stimulation at all positions at the same time 38.0%, $p < 0.001$, two-way repeated measures ANOVA with post hoc Bonferroni).

We then asked whether the three targeted cortical regions were responsible for particular aspects of the reaching movement or to

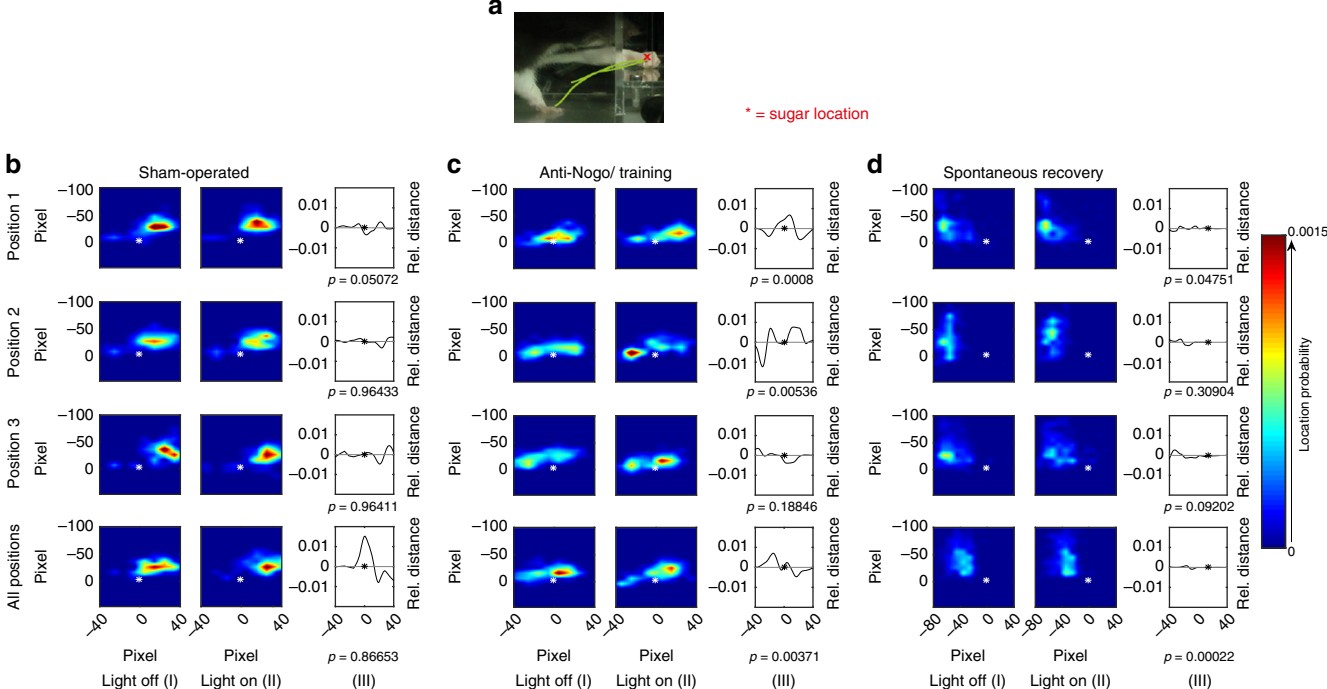

\* = sugar location

**Fig. 6** Animals with excellent motor recovery show specific regional organization of distinct microcircuitry in the contralesional cortex for the targeting kinematics of skilled grasping. **a** Side view (cutout) of a rat performing single pellet grasping. The paw trajectory of a successful grasp is superimposed. The furthest extension of the paw, location *x*, was automatically detected to capture the grasping trajectory in relation to the position of the sugar pellet. **b–d** Spatial distribution of the furthest extension calculated using every grasping trial under "light-off" (I) and "light-on" (II) conditions; (III) relative distance between "light-off" and "light-on" (\* = position of the sugar pellet). Therefore, the spatial distribution (I) and (II) are marginalized onto the *x*-location, before subtracting the resulting one-dimensional distributions to obtain (III). **b** Targeting of the paw to its final position over the sugar pellet in "sham-operated" animals was unaffected by light stimulation at positions 1–3 of the sensorimotor cortex independently or by concurrent light exposure of all cortical positions ("all positions": positions 1–3 together). **c** Animals of the "Anti-Nogo/Training" group with excellent recovery of grasping reached too far when the recrossed corticospinal neurons were inhibited in the contralesional premotor cortex position 1 or too short when the neurons in M1 (pos. 2) were silenced, whereas silencing close to S1 (Fig. 5d, pos. 3) did not affect the grasping behavior under "light-on" conditions. **d** Paw targeting of the "Spontaneous recovery" animals was poor overall and only significantly altered when all three cortical positions were optogenetically silenced at the same time (all position)

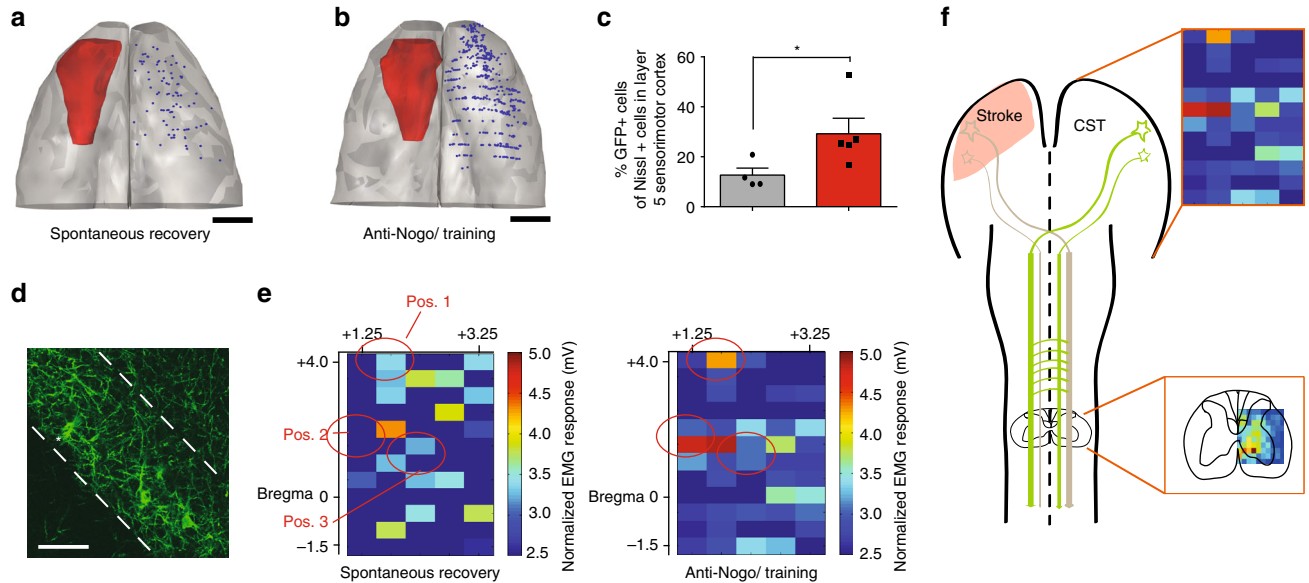

**Fig. 7** In animals with excellent recovery more contralesional corticospinal neurons project to the stroke denervated cervical hemi-cord and are clustered in M1 and premotor cortex. **a**, **b** Representative neurolucida reconstructions of localization and amount of corticospinal neurons in layer 5 expressing GFP, induced by the retrograde transport of AAV2.9-CamKII-Cre from the stroke denervated hemi-cord (s. Fig. 5b) in an animal of the "Spontaneous recovery" group (**a**) and one of the "Anti-Nogo/Training" group (**b**), scale bars = 2 mm. **c** GFP-positive neurons as percent of Nissl-positive cells in layer 5 of the premotor and motor cortex (4.2 mm ± 500 μm anterior and 0.67 mm ± 300 μm posterior to bregma): Significant more GFP-positive cells were found in anti-Nogo animals (red) compared to animals with spontaneous recovery (gray column, $p > 0.05$, paired $t$-test, with post hoc Bonferroni). **d** Examples of GFP-positive corticospinal-projecting pyramidal cells in layer 5 in the sensorimotor cortex (scale bar = 40 μm). **e** Intracortical microstimulation reveals "hot spots" in M1 and premotor cortex (M2) for circuits that evoke ipsilateral EMG responses from the contralesional M1 and M2 in "Anti-Nogo/Training" animals ($n = 4$). In contrast, in the "Spontaneous recovery" group ($n = 3$), only a diffuse cortical pattern for the evocation of EMG responses in the stroke-impaired forelimb was found. Heat maps of the cortical stimulation grid (60 stimulation points, 80 μA, + 4 to −1.5 mm AP and 1.25 to 3.25 mm ML relative to bregma) are shown whereby each stimulation point is color coded with the mean value of EMG response in mV (mean of EMG amplitudes in wrist, elbow, and shoulder) for the stroke-impaired forelimb. The red circles indicate the previous positions of the three optical implants. EMG responses are significantly pronounced in "Anti-Nogo/Training" animals at Pos. 2 compared to the "Spontaneous recovery" group (Pos. 1 $p = 0.149$; Pos. 2 $p = 0.0013$; Pos. 3 $p = > 0.99$, Mann–Whitney test). **f** Scheme: Our results suggest not only localized functional reorganization in the contralesional sensorimotor cortex, but also sprouting of midline crossing and ipsilaterally projecting CST fibers in lamina 6/7 of the stroke denervated cervical hemi-cord. Asterisks indicate significances: *$p < 0.05$

specific movement components. We used machine learning algorithms to dissect different aspects of the grasping kinematics and to analyze how they were affected by the "light-on" and "light-off" situation. Upon silencing of the rewired CST fibers in the three cortical areas, we detected distinct differences of the forepaw targeting, especially with regard to the final paw position relative to the pellet (Fig. 6a): While optical silencing in neither of the three cortical areas in sham-operated animals altered the grasping performance (Fig. 6b), inhibiting corticospinal-projecting neurons by green light stimulation in contralesional forelimb motor cortex (position #2) in animals with excellent motor recovery ("Anti-Nogo/Training" group) resulted in too short grasping actions thus missing the pellet (Fig. 6c, $p = 0.0053$, light-off vs. light-on condition, K–S test, Supplementary Table 1). Silencing the premotor cortex (position #1) (Fig. 6c, $p = 0.0008$, K–S test) induced too far grasping behavior (Supplementary Table 1). The same effect also occurred slightly weaker when inhibiting all positions (Fig. 6c, $p = 0.003$, K–S test, Supplementary Table 1). In the "Spontaneous recovery" group, significant differences were found upon light stimulation at position #1 and all three positions (Fig. 6d), but targeting was already strongly impaired during the "light-off" situation, so that additional light inhibition caused only mild, further perturbations.

We then counted and localized all ArchT-GFP-expressing corticospinal-projecting neurons in the contralesional hemisphere (Fig. 7a–d). Corresponding to the higher number of midline crossing corticospinal fibers in the denervated cervical spinal cord

in the "Anti-Nogo/Training" animals (Fig. 4a, b), we identified a significantly higher number of GFP-positive neurons in the contralesional motor cortex for the "Anti-Nogo/Training" group compared to the "Spontaneous recovery" animals group (Fig. 7a–c, for C $p < 0.05$, paired $t$-test, with post hoc Bonferroni).

Finally, we used intracortical microstimulation (ICMS) to confirm the localization of corticospinal neurons projecting from the contralesional hemisphere to the ipsilateral stroke denervated forelimb hemi-cord. We used a $5 \times 12$ point stimulation grid (positioned at +4 to −1.5 mm anterio-posterior and 1.25–3.75 mm medio-lateral coordinates relative to bregma) and electromyogram (EMG) recordings of wrist, elbow, and shoulder muscles of the impaired paw as readouts (Fig. 7e). We found a diffuse response pattern upon cortical electrical stimulation all over the contralesional pre- and primary motor cortex in animals without rehabilitative treatment (Fig. 7e, "Spontaneous recovery" group). In contrast, clear cortical "hot spots" in M1 and the rostral forelimb field that elicited motor responses in the formerly impaired forelimb, in particular for wrist and elbow, were detectable in the animals of the "Anti-Nogo/Training" group (Fig. 7e and Supplementary Fig. 6), indicating a highly localized representation of the reorganized corticospinal neurons projecting to the ipsilateral forelimb. The M1 ipsilateral forelimb "hotspot" identified by ICMS corresponded well to the position 2 (M1) of the optical implants for the silencing experiments described above. The EMG responses evoked around position 2 were significantly higher in "Anti-Nogo/Training" animals than

in animals with spontaneous recovery ($p < 0.05$, Mann–Whitney test). Our results suggest not only a distinct sprouting pattern of CST fibers from the contralesional hemisphere into the ventral horn of the denervated cervical hemi-cord, but also provide hints for a regionalized functional reorganization in the contralesional M1 and M2 (Fig. 7f).

## Discussion

We showed that short periods of direct and selective optogenetic stimulation of the intact corticospinal tract result in robust recovery of skilled forelimb functions after a large stroke, which completely destroyed the ipsilesional motor cortex, in particular if combined with subsequent intensive rehabilitative training. The stimulation induced corticospinal sprouting and axonal growth into the denervated cervical hemi-cord. Surprisingly, the stimulated stroke animals regained close to 100% of their original skilled forelimb movement abilities, and the kinematic analysis suggested a true recovery of the grasping behavior rather than compensatory movements. Selective optogenetic silencing of the rewired ipsilaterally projecting corticospinal fibers showed the crucial functional role of these fibers as well as differential effects of their cells of origin (related to reaching distance) in premotor and M1 cortices on the restored grasping function.

The role of activity as an important factor for fiber growth and neuronal rewiring after lesions is well recognized. But it has remained unclear whether growth, growth arrest and target interaction, synaptogenesis, or stabilization/pruning are the main ways by which activity drives the recovery processes[29–33]. In stroke, existing and newly formed connections of the contralateral cortex to the denervated targets could be strengthened by long-term potentiation such as the application of tetanic stimulation trains at 100 Hz[34, 35]. Here we selectively stimulated the motor cortex output neurons to the spinal cord on the intact, contralesional side of the brain at an early stage after the injury (d 3–14) with three daily short periods of $3 \times 1$ min stimuli. Earlier experimental studies stimulated the cortex electrically up to several hours daily at threshold levels[18, 21]. In a recent study in mice with 3 min daily stimulation of the ipsilesional motor cortex, upregulation of several neuronal growth and plasticity-promoting factors were seen in the contralesional M1[8]. These and the present data suggest that early brief periods of specific cortical and in particular corticospinal stimulation after stroke increase the capacity for reparative fiber growth and plasticity of the adult cortex. We found that optogenetic stimulation in combination with training robustly induces CST sprouting of pre-existing ipsilaterally projecting axons as well as midline crossing CST fibers in the denervated cervical hemi-cord. Furthermore, increased activity caused by repetitive optogenetic stimulation may have also enhanced corticospinal neurons to sprout into other brain regions such as the red nucleus and other brain stem areas (reticular formation, raphe nucleus) as previously described[36–38]. The here presented concept has direct clinical relevance; transcranial magnetic stimulation (TMS) and transcranial direct current stimulation (tDCS) are performed in stroke patients and were reported to enhance functional recovery. In some studies, mostly short-term effects were analyzed[39], but others suggested longer or even permanent beneficial effects of the stimulations[40–43]. More detailed animal and human studies are needed, in particular also with regard to the specific time windows and stimulation parameters, but the fact that repetitive, brief stimulations of corticospinal neurons can greatly enhance their sprouting, rewiring and functional recovery is encouraging for future stroke therapies.

We compared the effect of 2 weeks of optogenetic stimulation of the intact corticospinal tract followed by 2 weeks of intensive grasping training to an already established rehabilitative schedule with the same time intervals: Two weeks of Anti-Nogo immunotherapy, promoting axonal fiber growth, followed by 2 weeks of intensive grasping training[22]. The results of both these procedures were very similar: enhanced sprouting of corticospinal fibers from the intact contralesional cortex, reinnervation of the denervated cervical hemi-cord, and full restoration of impaired grasping function analogous to baseline levels. This equivalence supports the hypothesis of a growth-promoting effect induced by the stimulation paradigm used.

A key question in rehabilitation is whether a therapeutic intervention yields accurate restoration of lost movement patterns or only a compensatory movement strategy[44]. We developed an automated, unsupervised computer vision algorithm for detailed analysis of paw posture and kinematics during single pellet grasping. The algorithm uses routine high-resolution video sequences; no tattoos or manual tracing of joint positions are required. Conventional methods include manual analysis such as the 10-point evaluation suggested by Whishaw[45], which is time consuming and often subjective, or kinematic analysis of grasping trajectories[46]. However, pure grasping trajectories are not sensitive enough to detect delicate disparity of distinct aspects of the grasping act such as supination, targeting, or paw closure[22]. In contrast, our method combines a single frame analysis with a sequence matching approach, meaning that each recorded image of a grasping sequence is ranked for its temporal evolution of paw shape. Our algorithm thus enables to classify each grasp as closer or further away from previously identified typical healthy grasping sequences. Using this objective evaluation technique, we found that precision in targeting and the typical sequence of postures for successful grasping movements have recovered at the end of the experimental therapies (4–5 weeks after stroke) for those animals, which had either received direct optogenetic stimulation of the intact CST or Anti-Nogo immunotherapy followed by intensive grasping. This result shows a full and true restoration of forelimb function in rats after a > 95% destruction of one sensory-motor cortex and treated with growth-promoting therapies followed by intense rehabilitative training.

We implanted optical fibers for optogenetic stimulation locally over the premotor cortex, the main M1 cortex and the lateral motor and S1 cortex, and combined it with retrogradely transported virus that allowed selective expression of the optogenetic silencing construct ArchT in the rewired, ipsilaterally projecting corticospinal neurons. We were thus able to analyze the specific functional role of these three cortical subregions during the grasping. In the "Spontaneous recovery" group, pellet reaching remained severely impaired; optogenetic silencing concurrently at all positions caused a mild further impairment. Very much in contrast, the "Anti-Nogo/Training" group, which showed a full recovery of skilled reaching, reacted to silencing of the premotor cortex by overshooting reach movements, and by too short reaches after silencing of M1. When silencing at all three positions at the same time, animals still showed too long grasps but less significant compared to the silencing of the central M1, which may even suggest a more dominant role of the premotor cortex for the targeting aspect of the grasping than for M1. Another aspect may be the uneven distribution of rewired corticospinal-projecting fibers with a higher cell density expressing ArchT in the premotor and motor cortex vs. S1 (Fig. 7b), which may also have influenced the magnitude of the optogenetic silencing effect in the different positions (Fig. 5c) in the "Anti-Nogo/Training" group. In these functionally well recovered rats, we found "hot spots" for forelimb movements using ICMS cortical mapping in similar positions where silencing had decreased the grasping function. In contrast, the positions from which ipsilateral forelimb movements could be elicited were more diffusely distributed

over M1 and M2 in animals with a lower level of recovery. Also, a significant reduced cell density expressing ArchT had been detected in the "Spontaneous recovery" group, which may furthermore contribute to the reduced effect of optogenetic silencing in those animals. Cortical motor maps as the basis of voluntary movements have been described extensively: In particular, electrophysiological studies in monkeys, cats, and rodents have detected reproducible, complex movements of the forelimbs and hindlimbs triggered from distinct positions of the motor cortex[47–49]: Harrison et al., e.g., demonstrated that the forelimb motor cortex is subdivided in functional subregions for abduction and adduction movements, while others report that motor cortex silencing in the intact rodent interferes with the initiation or performance of trained fine motor tasks[50, 51]. In mice with forelimb motor cortex strokes, secondary-induced strokes in the premotor cortex destroyed regained skilled grasping function suggesting an important role of M2 for the regain of lost forelimb function[52]. However, the ability of the contralesional pre- and motor cortex to induce local circuitry of distinct aspects for fine motor function after stroke has been not revealed before. We show here for the first time that after an injury such as stroke and a successful rehabilitative intervention, plastic, rewired neurons, —and presumably entire microcircuits,—are found in the corresponding anatomical positions and are involved in specific aspects of the grasping sequence. Furthermore, this suggests that neurons retain much of their anatomical and functional identity while switching their axonal projection from one side of the spinal cord to the other. Such mechanisms could explain the common clinical and experimental observation that task-specific training often achieves much higher levels of success and functional restitution than generalized, multitask training paradigms[53–56].

What could be the neurobiological basis of the successful combination of direct, early cortical stimulation or a growth-promoting pharmacological treatment followed by intensive rehabilitative training? Both rehabilitation groups, the "Anti-Nogo/Training" and the "OptoStim/Training" combined an early, plasticity-promoting treatment approach followed by intensive rehabilitative training of the impaired paw. This sequential combination resulted in almost full recovery of impaired motor function. In addition, for both treatment groups a similar sprouting pattern of CST fibers targeting motor neuronal pools in lamina 6/7 of the ventral horn was found, suggesting at least an additive effect of the two therapies: First the stimulation of circuit formation and reorganization followed by use-dependent selection, stabilization, and strengthening of meaningful circuitry by rehabilitative training. In stroke patients, new circuit formation can be enhanced beyond the intrinsic plastic potential of the brain by growth stimulatory therapies applied early after stroke, followed by a step of circuit selection and stabilization by intensive rehabilitative training to enable specific functional shaping of new circuits to restore the lost motor functions. That brief bouts of targeted cortical motoneuronal stimulation early after stroke can enhance neuronal growth and repair is a new finding which can be translated into clinical trials using established clinical techniques such as TMS or tDCS.

## Methods

**Animals.** Subjects of this study were a total of $n = 46$ adult female Long-Evans rats (200–250 g, 3–4 months of age, Janvier, France), housed in groups of two to four under a constant 12 h dark/light cycle with food and water ad libitum. All experimental procedures were approved by the veterinary office of the canton of Zurich, Switzerland. They are in accordance with the Stroke Therapy Academic Industry Roundtable (STAIR) criteria[57] for preclinical stroke investigations.

**Experimental outline.** The objective of this study was to induce sprouting of corticospinal fibers from the intact hemisphere to the denervated cervical spinal

cord and thus promote recovery of skilled motor function by distinct optogenetic stimulation of the intact corticospinal tract after a large photothrombotic stroke. Rats were first handled and training in the single pellet grasping task to determine their paw preference. Depending on their left or right handedness, a retrograde AAV9-CamKII0.4.Cre.SV40 vector (Penn Vector Core, Philadelphia) was injected in the contralateral cervical hemi-spinal cord of the preferred paw (contralateral to the "future" denervated cervical hemi-spinal cord) followed by the injection of a Cre-recombinase-dependent ChR2 vector (AAV2.1_Ef1a-DIO-hChR2(t159C)-mCherry, UNC, Chapel Hill, USA) in the ipsilateral pre- and sensorimotor cortex (in the "future" contralesional hemisphere) thus achieving the specific expression of ChR in corticospinal-projecting neurons. Two weeks after virus injection, three optic fiber implants were positioned over the pre- and sensorimotor cortex, where the viruses had been injected. After recovery from the surgeries, the training in the single pellet grasping task was continued till a baseline level of at least 60% success rate in the fine motor task was achieved. All rats then received a photothrombotic stroke targeting the sensorimotor cortex corresponding to their paw preference in the grasping task (contralateral to the hemisphere of the fiber implantation). Two days after stroke, animals were retested in the single pellet grasping task and according to their lesion deficit randomized in four different rehabilitation groups (Fig. 1a): In the "Spontaneous recovery" group, animals received no rehabilitative treatment and were only assessed for regain of grasping function weekly up to 4–5 weeks after stroke. The intact pre- and sensorimotor cortex of animals in the "OptoStim" group was stimulated with blue light 3×/day within the first 2 weeks after stroke to activate the ChR-expressing corticospinal-projecting neurons. Additionally to the optical stimulation, animals in the "OptoStim/Training" group underwent intensive grasping training of the impaired paw during the 3rd and 4th week after stroke. Animals in the "Delayed Training" group were also intensively trained in the single pellet grasping task during the 3rd and 4th week after stroke but without optical stimulation in advance. After the completion of the rehabilitative schedules, animals performed novel tasks to assess their overall recovery of motor function, followed by the removal of the optical fibers and anterograde tracing of the intact pre- and sensorimotor cortex with BDA. This experimental setup was repeated in $N = 2$ independent studies with $n = 8$–12 rats per cohort for behavioral and morphological analysis. As there was no statistically significant difference in the outcome of lesions, behavior and anatomy, the data shown here were pooled from both studies. The final behavioral and anatomical analysis was performed by an independent investigator who was not involved in the rehabilitative training and testing. During the experiments for Fig. 1, rehabilitative training was twice perturbed on days 15 and 16: Once by construction works and then by a change of the operational procedure of the animal facility so that the data gained from days 15 and 16 after stroke were inconsistent (Fig. 2a) and thus not included in the final analysis. This study confirms with the AARIVE guidelines (https://www.nc3rs.org.uk/arrive-guidelines).

In order to compare the efficacy of a rehabilitative scheme with direct optogenetic stimulation of the intact corticospinal tract, we applied a rehabilitative schedule including a growth-promoting immunotherapy and rehabilitative training to a second cohort of animals, which has been shown to induce sprouting of corticospinal-projecting fibers from the intact motor cortex after stroke[22]: All animals were trained in single pellet grasping and were then divided in two groups: While a part of animals was sham-operated, the other animals received a photothrombotic stroke destroying the sensorimotor cortex of the preferred paw. Two days after stroke, animals were randomly distributed in two groups: Either animals received no rehabilitative intervention ("Spontaneous recovery" group) or anti-Nogo-A antibodies were applied for 2 weeks immediately after stroke followed by grasping training for another 2 weeks (Fig. 2a).

**Photothrombotic stroke.** A photothrombotic stroke was introduced as previously described[9, 22]: Animals were anesthetized with 3% isoflurane followed by an intramuscular injection of Medetomidin (30 μg/kg body weight), Midazolam (0.4 mg/kg body weight), and Fentanyl (1 μg/kg body weight). The rats were shaved, fixed in a stereotactic frame, eye cream (Vitamin A, Braun) was applied and the scalp was opened after disinfection (Betadine, Braun). Two minutes after an i.v. injection of Rose Bengal into the femoral vein (Sigma, 10 mg/ml solved in 0.9% NaCl solution) according to the body weight (13 mg/kg) a 10 × 5 mm area (localized 5 mm to −5 mm anterior and 0.5 mm to 5.5 mm lateral to Bregma) was exposed to a strong light source (Olympus KL 1500LCS, 150 W, 3000 K) for 12 min illumination through the intact scalp. To antagonize the anesthesia Antisedan (0.15 mg/kg body weight) and Flumazenil (40 μg/kg body weight) were s.c. injected. For postoperative care, all animals received analgesics (Dalfalgan Sirup, Braun, per os in the drinking water) and antibiotics (Baytril, 5 mg/kg body weight, Bayer, s.c.) for at least 3 days after surgery.

**Optogenetics.** We used a Cre-dependent approach to specifically express ChR2 in the corticospinal tract requiring two surgeries: For the first surgery, animals were anesthetized as mentioned previously for the stroke lesion and a minimal invasive laminectomy at spinal level C5–C6 was performed. A total of 11 × 120 nl of the retrograde AAV9-CamKII0.4.Cre.SV40 vector (Penn Vector Core, Philadelphia) was injected in the cervical hemi-spinal cord (Fig. 1b), contralaterally to the future denervated hemi-spinal cord due to the stroke. The 11 injections took place 0.7 mm lateral to the midline using a 35-gage, 10 μl syringe (Hamilton, BGB Analytik)

driven by an electric pump (World Precision Instruments) with a flow rate of 6 nl/s. Each injection was made in 2 steps of 60 nl and 2 different depths (0.9 and 1.2 mm below the spinal cord surface to target the corticospinal tract and lamina 7 motor neuron pools) keeping the syringe in place for 3 min between steps. One–two days after laminectomy, a craniotomy was performed exposing the future contralesional pre- and sensorimotor cortex: 12 injections (200 nl each) of a Cre-recombinase-dependent ChR2 vector (AAV2.1-Ef1a-DIO-hChR2(t159C)-mCherry, UNC, Chapel Hill, USA) were made through the intact dura using a 33-gage, 10 μl syringe (World Precision instruments) with a flow rate of 6 nl/s controlled by an electrical pump (World Precision instruments). The following injection coordinates were used: 2 mm anterior to bregma (AP), 3 mm lateral to bregma (ML); 2.5 AP, 3.5 ML; 1.5 AP, 3.5 ML; 1.5 AP, 2.5 ML; 2.5 AP, 2.5 ML; 3.5 AP, 2.0 ML; 3.5 AP, 2.5 ML; 4.5 AP, 2.5 ML; 4.5 AP, 1.5 ML; 2.0 AP, 1.5 ML; 2.0 AP, 1.5 ML; 4.0 AP, 0.15 ML. At each position, the needle was lowered to 1.5 mm depth and retracted to 1.2 mm again to leave a pocket for the volume injected. The needle was left in place for 1 min before 300 nl were administered. On injection completion, the cannula was left to rest for another 2 min before full retraction to ensure proper diffusion of the dispatched virus. The craniotomy was covered with Kwik-Cast (World precision instruments) and the scalp was sutured till reopening for the implantation of the optical fibers.

Custom-made optic glass fibers (multimode fibers, 0.48 NA, High OH, Ø 1000 μm Core, Thorlabs) glued into stainless steel sleeves with two-component epoxy resin glue and sanded (Silicon Carbid Lapping Sheet, 1, 3, and 5 um grit, Thorlabs) were used as the medium for light conductance. Only implants with a conductance above 70% were included (mean conductance of the used implants was 73 ± 4%) as measured by a power meter (473 nm, Thorlabs). For implantation of these optical fibers, animals were anesthetized as previously described for the photothrombotic stroke and a single subcutaneous injection of mannitol (20%, 17 mg/ml, B. Braun) was applied to reduce swelling of the cortex. Soft tissue was removed from the skull, iBond (Heraeus) was deposited and four screws were inserted around the craniotomy for the fixation of the optical implants. The three optical implants were positioned on the surface of the pre- and sensorimotor cortex according to the following coordinates: Optical implant #1 + 4.0 mm anterior to bregma (AP), +2.0 mm lateral to bregma (ML); optical implant #2 + 2.5 AP, +1.5 ML; optical implant #3 + 1.5 AP, +2.5 ML, external angle 0.9° (Fig. 5d and Supplementary Fig. 2). A thin layer of transparent silicon (Kwik-Sil, World Precision Instruments) covered the surface around the implants. All implants were fixed to the skull via the screws using light hardened dental cement (Tetric EvoFlow, Ivoclar vivadent). The scalp was then sutured and glued to the cemented basis with 3 M Vetbond™ (3 M Animal Care Products, USA). Baytril (5 mg/kg body weight, Bayer) was subcutaneously applied for the first 3 days after stroke followed by per os administration up to 1–2 weeks after surgery (190 μl/250 ml drinking water).

For optical stimulation of the intact corticospinal tract, three blue LEDs (473 nm, Thorlabs) were adjusted to provide the same output on full input strength (output difference of less than 2% ($64 \pm 1$ mW/mm$^2$) as measured by a power meter (473 nm, Thorlabs)). For the illumination during the stimulation therapy, the output at the tip of each individual optical implant was adjusted to 16.6 mW/mm$^2$. Animals in the "OptoStim" group and "OptoStim/Training" group received optical stimulation of the intact corticospinal tract 3×/day from day 3 to day 14 after stroke. Each stimulation session consisted of $3 \times 1$-min stimulation with 10 Hz, 20 ms pulses. Between the 1-min stimulation phases were 3-min resting phases as previously described[8]. The pulses were synchronized over all LEDs and thus could irradiate the cortex through all three implants simultaneously. For the stimulation, animals were put in a Plexiglas box where the optic fibers (Ø 1 mm), connected with ceramic sleeves (Thorlabs) to the optical implants were supported by a freely moving arm above the box designed to reduce the weight of the fibers while allowing the animals to move relatively unhindered.

**Rehabilitative training and testing.** All animals included were trained in the single pellet grasping task[5, 22] to assess fine motor control of the forelimb. Animals were placed in a Plexiglas box ($34 \times 14$ cm) with two openings on opposite ends and were trained to grasp pellets (45-mg dustless precision pellets, TSE Systems Intl. Group). Grasping performance was scored as follows[22]: a grasp was classified of being successful (scored as "1") if the animal correctly targeted a newly presented pellet with its preferred paw, retrieved it and brought it directly to its mouth. A score of 1 was also given if the animal required several attempts to grasp the pellet, without retracting the paw through the window and into the box, which was defined as the end of an attempt. A score of 0.5 was given if—after a successful grasp—the pellet was dropped inside the box. If the animal knocked the pellet off the shelf, the trial was scored as 0. The success rate was calculated as the percentage of retrieved pellets of the number of all trials. Animals were trained 3–4 weeks for baseline recordings before stroke. Only animals with a 60% or higher success rate at baseline were included in the study presented here. During testing sessions, animals were given 20 pellets within a maximum time of 10 min. During rehabilitative training sessions, animals grasped at least 100 pellets/day. All animals were assessed at baseline as well as 2 days, 7 days, and until 4–5 weeks after stroke for skilled forelimb function in the single pellet grasping task. All sessions were filmed (Panasonic HDC-SD800 High Definition Camcorder, 50 frames/s) for further analysis of grasping kinematics. The researchers responsible of the development of the computer code for the behavior analysis were blinded to all behavior experiments presented in this study.

**Computer vision algorithms for posture analysis and analysis of grasping kinematics.** We aimed at developing a non-invasive approach for analyzing motor function that does not interfere with the recovery process of the animal in order to monitor the progress of therapeutic treatment and to identify the optimal treatment protocol. Therefore, we here present an automatic, purely visual analysis based on video recordings of the grasping animals. A detailed investigation of not only the spatial trajectory of a grasp, but also of paw posture and its change during grasping then revealed even subtle differences between impaired and healthy motor function. Therefore, we needed to first detect and track fast moving paws under adverse lighting conditions (considering the speed of the paw, light is very dim so as not to irritate the rats) and reflections of the surrounding cage. Second, paw posture had to be represented in a robust manner. Third, a sequence matching approach was employed to model the characteristic changes of posture during grasping.

To focus our analysis on paw motor function, we first extracted a large set of candidate foreground regions. Following Wright et al., we decomposed video frames into a low-rank background model and a sparse vector corresponding to foreground pixels. Candidate regions $x_i \in \mathcal{X}$ are then randomly sampled from the estimated foreground of a set of sample videos to initialize the subsequent learning of classifiers and compute their HOG features (size $10 \times 10$)[58]. We also investigated features learned using a convolutional neural network approach[59]. However, due to the limited amount of training samples, standard HOG features yielded better performance and interpretable results. K-nearest neighbors density estimation then reveals rare outliers, which are removed from $\mathcal{X}$.

Max-projection of randomized exemplar classifiers: For paw detection and posture comparison, a set of exemplar classifiers were required, which detected paw configurations, that were similar to a set of prototypical ones[60]. To measure the similarity to a sampled region $x_i$, we trained a discriminative exemplar classifier $w_i$, with $x_i$ as positive example. Since $\mathcal{X}$ is likely to contain other samples similar to $x_i$, this set could not be used as negatives since too much overlap with the single positive exemplar would have corrupted learning. Instead of training a single classifier, we thus trained an ensemble of $K$ randomized exemplar classifiers $w_k$ with offsets $b_i^k$, $k \in \{1, \ldots, K\}$, using randomly selected negative sets $\mathcal{X}_i^k \subset \mathcal{X} \backslash x_i$,

$$\min_{w_i^k, b_i^k} \|w_i^k\|^2 + C \max\left(0, 1\langle w_i^k, x_i \rangle - b_i^k\right) + \frac{C}{|X_i^k|} \sum_{j=1}^{|X_i^k|} \max\left(0, 1 + \langle w_i^k, x_j \rangle + b_i^k\right).$$

Cross-validation yielded a soft margin $C = 0.01$. To compensate for the unreliability of individual classifiers, they were aggregated using a max-projection which chose for each feature dimension the most confident classifier $w_i := \max_k w_i^k$. The scores of these max-projected $w_i$ were then calibrated by a logistic regression[60].

Codebook of prototypical paw classifiers: Based on the large initial candidate set $\mathcal{X}$ (size 1000), we then created a dictionary $\mathcal{D} \subseteq \mathcal{X}$ as a canonically representation of paw postures of previously unseen rats. The randomized exemplar classifiers provided a robust measure of pair-wise similarity $s(x_i, x_j) := \frac{1}{2}\left(\langle w_i, x_j \rangle + \langle w_j, x_i \rangle\right)$. Duplicate candidates were merged using normalized cuts (Shi and Malik 1997) to obtain a dictionary $\mathcal{D}$ (size 100), which was sufficiently diverse and rich for a non-parametric representation of all paw postures. In a new video, all classifiers $w_i$ from dictionary $\mathcal{D}$ were applied densely. We averaged the scores of the top scoring $k$ classifiers and took the location with the highest score to obtain the final paw detection and its trajectories over time.

Non-parametric description of paw posture: On new paw detections $x$, the activation pattern of all $\{w_i\}_{i \in \mathcal{D}}$ gave rise to an embedding $e := [\langle w_1, x \rangle, \ldots, \langle w_{|\mathcal{D}|}, x \rangle]$. Moreover, novel paw configurations could now be compared using their embedding vectors, since similar samples yielded similar classifier activations. In Fig. 3, left column, we visualized paw postures by mapping the high-dimensional embedding to a low-dimensional projection using the t-SNE method[26].

Spatiotemporal parsing: While paw posture was informative, its deformation during a grasp was even more characteristic. Modeling not only the grasping behavior of healthy specimen, but also the diverse abnormal patterns after impairment rendered prior models on posture and kinematics infeasible, as they would impose a bias toward healthy configurations. We represented an $M$ frame long grasp $j$ as a sequence in the embedding space $S_j := [e_1^j, \ldots, e_M^j]$. Measuring the similarity of grasps based on their embeddings depended on a sequence matching since they were not temporally aligned. We thus searched for a mapping $\pi$ : $\{1, \ldots, M\} \mapsto \{0, 1, \ldots, M'\}$, which aligned $S_j$ of length $M$ with $S_{j'}$ of length $M'$ (0 were outliers). We then defined their distance in the embedding space as $d(S_j, S_{j'})$ : $= \sum_{i=1}^{M} |e_i^j - e_{\pi(i)}^{j'}|$. A matching $\pi(\bullet)$ should avoid outliers and changes of the temporal order,

$$\min_{\pi} \sum_{i=1}^{M} \left\| e_i^j - e_{\pi(i)}^{j'} \right\| + \lambda \sum_{i=1}^{M-1} 1(\pi(i) > \pi(i+1)), \text{ s.t. } |\pi(i) - i| \leq B, \forall i,$$

where $1(\cdot)$ was the identity function. The constraint in this equation prevented matched frames to from being more than $B$ frames apart from another. The second

sum allowed for more flexible matching than in standard string matching or dynamic time warping. The sequence matching problem was then solved using integer linear programming and IBM ILOG CPLEX. As it would be computationally prohibitive to match all sequences against another, we eliminated redundancies by constructing a dictionary $\mathcal{D}_{seq} := \{S_1, \ldots, S_Q\}$ of canonical grasping sequences, which then explained novel grasps yielding a spatiotemporal parsing. Measuring the distances of a new sequence $S'$ to all prototypical ones in $\mathcal{D}_{seq}$ produced the sequence-level embedding $F' := [d(S', S_1), \ldots, d(S', S_Q)]$, after aligning grasps with our sequence matching. Grasps are then compared using the Euclidean distance in this sequence embedding space. In Fig. 2c, we illustrate the similarity of grasping trials to baseline and 2d sequences per cohort using the previously described approach. We furthermore utilize the sequence-level embedding for evaluating the initial learning of the skilled reaching task. Supplementary Fig. 3 illustrates the learning behavior of rodents and demonstrates the high sensitivity of our algorithm, which even detects small improvements/changes in grasping during the first days of training.

**Novel tasks.** After the completion of the rehabilitative schedules (4 weeks after stroke), animals were exposed to novel tasks of forelimb sensorimotor function and overall locomotion such as the horizontal ladder test[61] and the narrow beam task[62] in order to determine their non-task-specific recovery levels of forelimb function.

The irregular horizontal ladder crossing was performed as described recently[22]: On three consecutive days, three runs per animal were recorded (Panasonic HDC-SD800 High Definition Camcorder) and analyzed frame by frame (VideoReDo TV Suite, H 264, Drd Systems Inc.). The success rate was calculated by dividing the amount of correct steps by the number of total steps taken with the respective paw ×100 in the three test runs.

For the narrow beam task, rats were placed at one end of a wooden 145 cm long, elevated beam with a basis of 5 cm width narrowing down at the end tip to a width of 1 cm with 28 equal segments of 5 cm. While the animals were crossing the beam, two investigators on both sides of the beam detected at which segment a slip of one of the extremities down from the plane surface of the beam would first occur. The success rate was calculated by dividing the segments noted for the first slip by the total amount of segments (×/28) × 100. For each day of assessment four different trials were scored and averaged per day. For both, the horizontal ladder and the narrow beam task animals were tested on 3 consecutive days and the success rates are given as mean of the three testing days.

**Anterograde tracing of the intact corticospinal tract (CST).** Six weeks after stroke, after the completion of the behavioral training and testing, the intact contralesional motor cortex of animals in all five rehabilitation groups ("OptoStim/Training" $n = 4$, "OptoStim" $n = 3$, "Spontaneous recovery" $n = 4$, "Delayed Training" $n = 4$, "Anti-Nogo/Training" $n = 6$) was traced anterogradely with Biotinylated Dextran Amine (BDA, 10,000 molecular weight, 10% solution in 0.01 M PBS, Invitrogen) as previously described[22]. Before the tracing, the optical fiber implants had to be removed to expose the contralesional motor cortex again. At this step, we excluded animals from further analysis if we found signs of inflammation or the removal of the cemented optical implants caused bleeding. Twelve injections (200 nl each) of BDA were made as described above for the expression of the ChR2 vector (AAV2.1_Ef1a-DIO-hChR2(t159C)-mCherry) using the same coordinates. Three weeks after BDA injections animals were anesthetized (pentobarbital, 450 mg/kg body weight i.p., Abbott Laboratories) and perfused transcardially with 100 ml Ringer solution (containing 100,000 IU/l heparin (Roche) and 0.25% NaNO₂) followed by 300 ml of a 4% phosphate-buffered paraformaldehyde solution, pH 7.4. Brains and spinal cords were dissolved and cryoprotected in a phosphate-buffered 30% sucrose solution for cryostat sectioning in 40 µm thick sections before being stained by on-slide processing using the nickel-enhanced DAB (3,3′-diaminobenzidine) protocol (Vectastain ABC Elite Kit, Vector Laboratories; 1:100 in Tris-buffered saline plus TritonTM X-100).

CST fiber growth in response to stroke was evaluated at spinal cord levels C3–Th1 as previously described[22]: Fibers crossing the spinal cord midline were counted at ×20 magnification and branching of these fibers was evaluated at four defined distances from the midline in the gray matter using a virtual grid (D1–D4), with each line 125 µm apart[22]. Ventral CST fibers were counted at the transition of the white matter forming the anterior median fissure and the gray matter of the ventral horn. A virtual line was aligned with the center of the central canal. Around this point, the line was rotated and aligned with the boundary between gray and white matter. Fibers crossing this line were counted as ventral CST fibers innervating the gray matter[9].

To correct for variations in BDA labeling, data were normalized to the number of BDA-labeled axons for each animal (CST axons: counted in five rectangular areas (4000 µm²) and extrapolated to the total area of the CST (0.1144 mm² ± 0.01)/slice in two sections at spinal cord level C3 and C6). Results are expressed as newly outsprouting fibers divided by the number of labeled fibers in the intact CST/slice for each animal.

**Histological analysis of stroke volume.** For analysis of stroke volume, animals were transcardially perfused and rat brains were fixed and cryoprotected as described in the section above. Brains were cut coronally on a cryostat in 40 µm sections and collected on slides (SuperFrost®) as described previously[22]. Frozen

sections were dried at room temperature, rehydrated and immersed in 0.5% cresyl violet (2 min) for Nissl staining. After washing in water, we dehydrated the sections in graded alcohols, cleared in xylene, and cover-slipped with Eukit® mounting medium. The lesion volume of all four rehabilitation groups was evaluated by 3D reconstruction of every 20th brain section, using Neurolucida 8.0 (MicroBright-Field). The lesion volume of each section was determined by contouring the destroyed tissue and taking also into account the symmetry of stroke-dependent tissue loss in relation to the volume of the intact contralesional hemisphere. The stroke lesion location was determined using the coordinates of the Rat Brain Atlas by Paxinos and Watson (Ref. [63] and Supplementary Fig. 1).

**Histological verification of ChR2-expressing neurons.** Coronal cortex sections (40 µm slice thickness) were examined for the distribution of mCherry-positive neurons with anti-mCherry amplification immunohistochemistry: Immediately after cryo-sectioning, slices were blocked in Tris-NaCl-blocking buffer and 0.1% Triton (TNB, TBST) for 30 min at room temperature and incubated over night at 4 °C in TNB, TBST, and the primary antibody mCherry (1:1000; Abcam, Lucernachem). After washing with 0.1 M phosphate buffer, the biotinylated goat anti-rat immunoglobulin-G (Ig G) (1:300; Jackson IR) secondary antibody in PBS was applied for 1 h at room temperature. The sections were washed in PBS and incubated in Streptavidin C3 (1:1000; Jackson IR) and fluorescent Nissl 500/525 (Neurotrace Green Fluorescent Nissl Stain, Invitrogen) in TNB for 1 h. Slices were washed and rinsed with 0.05 M Tris-buffer for the final step. Images were acquired with a confocal microscope (TCS SP2 AOBS, Leica) with respective red (Cy3) and green (488) excitation or emission filters at ×20 magnification. Images were taken in a sequential mode either for acquisition of mCherry- or Nissl-positive cells. The percentage of mCherry-positive cells of Nissl-positive cells in layer 5 of the primary motor cortex (4.2 mm ± 300 µm anterior and 0.67 mm ± 300 µm posterior to bregma, Fig. 1), premotor cortex (4.68 mm µm anterior to bregma) and primary sensory cortex (0.12 mm ± 300 µm anterior to bregma) was counted in five sections per animal and averaged per animal and group.

**Anti-Nogo immunotherapy.** To compare the effect of specific optical stimulation of the intact corticospinal tract to induce fiber sprouting in the cervical spinal cord, a group of animals ($n = 9$) received the growth-promoting anti-Nogo immunotherapy[9, 22]. For constant delivery of the function blocking Ig G1 mouse monoclonal anti-Nogo-A antibody 11C7 (3 or 4.2 mg/ml, Novartis) against an 18 amino acid Nogo-A peptide corresponding to the rat sequence amino acids 623–640[64] a fine intrathecal catheter (32 gage) was placed after stroke surgery in the subarachnoid space at lumbar level L2 after laminectomy and connected to an osmotic minipump (Alzet 2ML2; 5 µl/h, 3.1 µg/µl). For postoperative care, all animals received analgesics (Rimadyl, 2.5 mg/kg body weight, Pfeizer) and antibiotics (Baytril, 5 mg/kg body weight, Bayer) for 3 days as well as a single subcutaneous injection of mannitol (20%, 17 mg/ml, B. Braun) to reduce swelling of the cortex. Two weeks after stroke, pumps and catheters were removed.

**Optogenetic inhibition of newly outsprouting fibers of the intact corticospinal tract.** For optogenetic inhibition of newly outsprouting corticospinal-projecting fibers from the intact hemisphere, we applied a Cre-dependent approach: After completion of either 2 weeks of anti-Nogo immunotherapy followed by training ("Anti-Nogo/Training" group) or spontaneous recovery ("Spontaneous recovery" group) we injected 11 × 120 nl of the retrograde AAV9-CamKII0.4.Cre.SV40 vector (Penn Vector Core, Philadelphia) in the denervated cervical hemi-spinal cord (Fig. 5b) at spinal cord level C5–C6 as described above ("Expression of ChR2 in the intact corticospinal tract"). One–two days later, the Cre-dependent AAV2.1-Flex-ArchT-GFP (UNC GTC Vector Core, the University of North Carolina at Chapel Hill) was injected in the contralesional pre- and sensorimotor cortex according to the coordinates ("Expression of ChR2 in the intact corticospinal tract", Fig. 1b). Up to 7 days after virus injection, animals were implanted with three custom-made optic glass fibers at the same positions as described above ("Expression of ChR2 in the intact corticospinal tract") enabling inhibition at three distinct positions (position 1: pre- and rostral motor cortex; position 2: motor cortex; position 3: motor cortex with small parts of the primary sensory cortex). Sham-operated animals ($n = 3$) just received optical fiber implantation without virus injections to confirm that light stimulation and the fibers connecting the optical implants to the lasers did not influence grasping behavior.

**Optogenetic inhibition during a single pellet grasping task.** Animals were given 2 weeks of recovery from virus injection and fiber implantation surgeries and were then retrained and retested in the single pellet grasping task (Fig. 5a). For optical inhibition of distinct subsets of newly sprouting corticospinal fibers during the grasping task, animals were put in a Plexiglas box (section "Rehabilitative training and testing") with their optical implants connected via ceramic mating sleeves to 1 m long optical fibers (Ø400 µm Core, 0.39 NA, Thorlabs) and three lasers (532 nm) above the grasping box, enabling optical inhibition at each optical implant location independently. The three laser output strengths were adjusted to provide an output difference less than 2% (42.6 ± 1.7 mW/mm²). For the optical inhibition, the output strength was adapted to 20 mW/mm² for each individual laser using optical density filters. The initiation of the three lasers was regulated by

a light barrier positioned at the center of the provided food pellet so that by the first grasp through the light barrier a program (LabVIEW, National Instruments) was started which kept the lasers off for 100 s and turned them on afterwards for 100 s while the animal continuously grasped for pellets. After another lag time of 100 s with laser lights off the next grasp would initiate the second phase of laser light activation. We used continuous laser light stimulation for the ArchT activation, which has been shown to cause no apoptosis when lasers (532 nm) were turned on for even 3–5 min[65]. We only stimulated at either one position (position 1 vs. position 2 vs. position 3, Fig. 5c) or at all three positions at the same time during one grasping session. The light barrier controlled the light-off and -on phases at this position for several times as long as the animal continued with grasping for pellets. For the next session, we then randomly changed to stimulating at another position in order to avoid inhibiting the same area over several days of grasping behavior and thus inducing structural and plastic changes. The effect of optically inhibiting at a distinct position (1–3) or at all three positions on grasping success was measured for each animal and each position during at least three independent grasping sessions. Results are presented as mean success rates of all grasping sessions without light ("light-off phase") and for the optogenetic inhibition of a distinct position ("light-on phase") in absolute levels as well as relative success rates of the "light-on phases" in dependence of the success rates in the "light-off phases".

For the detailed kinematic and posture analysis of the grasping behavior under "light-off" and "light-on" conditions, we utilized the previously described detection and tracking of paws for a grasping trajectory analysis as well as the spatiotemporal parsing of grasping sequences for a detailed posture analysis that discovers even fine-grained differences between healthy and impaired motor function (please refer to "Computer vision algorithms for posture analysis and analysis of grasping kinematics").

**Histological verification of ArchT-expressing neurons**. Coronal cortical sections (40 μm slice thickness) were examined for the distribution of EGFP-positive neurons with anti-GFP immunohistochemistry similar to the one described above (section "Histological verification of ChR2-expressing neurons") except that for the primary antibody the primary antibody green fluorescent protein (GFP) (rat monoclonal antibody, 1:1000; Nacalai) was used as well as the fluorescent Nissl 640/660 (1:500; Neurotrace Invitrogen) in TNB for the Nissl staining. Analysis of GFP-positive cells in relation to Nissl-positive cells was undertaken as above (see section "Histological verification of ChR2 expressing neurons"). To localize the location of clusters of neurons expressing GFP 3D reconstructions of every 20th brain section was performed for a representative brain of each group ("Anti-Nogo/Training" and "Spontaneous recovery") using Neurolucida 8.0 (MicroBrightField).

**Intracortical microstimulation**. For intracortical microstimulation, "Anti-Nogo/Training" ($n = 4$) and "Spontaneous recovery" animals ($n = 3$) were anesthetized with a subcutaneous mixture of ketamine (50 mg/ml, 7 mg/kg body weight, Streuli Pharma) and xylazine (20 mg/ml, 5 mg/kg body weight, Streuli Pharma) plus a single injection of mannitol (20%, 17 ml/kg, B. Braun). The forelimbs were shaved for better visibility of muscles and the rat was mounted in a stereotactic frame[66]. The optical implants were removed exposing the entire pre- and sensorimotor cortex (5 mm to −2 mm AP, 4 mm lateral relative to bregma). Using bregma as a landmark, electrode penetrations were made perpendicular to the pial surface (depth 1.3 mm) tracing an rectangular exploration grid of 5 × 12 stimulation points localized from 4 mm to −1.5 mm AP and 1.25 to 3.25 mm ML relative to bregma with a distance of 500 μm for each stimulation point. Forty-five-millisecond trains of 0.2 ms biphasic pulses at 333 Hz[9, 22] with a current of 80 μA (insuring a stable response, ref. [20]) were delivered through a glass isolated platinum/tungsten stimulation electrode with an impedance of 0.5–1 MΩ (Thomas Recording). We used EMG recordings from the (ipsilateral) impaired forelimb (M. extensor digitorum for wrist, M. biceps and triceps for elbow, M. trapezius for shoulder) as readouts. The EMG signal was amplified, filtered, digitized, and visualized via PowerLab (AD instruments). The EMG data were subsequently transferred to Matlab and the maximum of the EMG-amplitude was detected at each stimulation point and for all joints. For each animal, we calculated the sum of the maximal amplitudes of all three joints for each stimulation point ($\mathrm{Sum_{Ampl}} = \mathrm{maxAmpl_{wirst}} + \mathrm{maxAmpl_{elbow}} + \mathrm{maxAmpl_{shoulder}}$). Heat maps were generated by calculating the median of $\mathrm{Sum_{Ampl}}$ for the animals of a group ("Anti-Nogo/Training" and "Spontaneous recovery" group) for each stimulation point representing median EMG responses of a group at a distinct location of cortical stimulation relative to bregma.

**Statistical analysis**. For statistical analysis, GraphPad Prism (GraphPad Software Inc.: Version 6.1) was used. All data are expressed as mean ± s.e.m. For comparing the behavioral and anatomical reorganization of the rehabilitation groups, a two-way ANOVA followed by Bonferroni's post hoc test was used. In all these experiments, differences between the rehabilitation groups or conditions ("light-off" vs. "light-on") were significant for the chosen size of the cohort, thus justifying samples size. Whenever two treatments were compared at one time point, Students t-test (paired, two-tailed) was used after checking both distributions of being Gaussian. To test for statistical significance in our kinematic analysis, a K–S test was used. The level of significance was set at *$p < 0.05$, **$p < 0.01$, and ***$p < 0.001$.

**Code availability**. The computer code describing the computational method of our behavior analysis is available at https://github.com/CompVis/AutomaticBehaviorAnalysis_NatureComm.

**Data availability**. The authors declare that the data supporting the findings of this study are available within the paper and its Supplementary Information files. The computer code and data together with a documentation describing the computational method of our behavior analysis as Supplementary Material are uploaded to github and are available at https://github.com/CompVis/AutomaticBehaviorAnalysis_NatureComm.

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

## Acknowledgements
We thank Dr Yaroslav Sych and Martin Wieckhorst for technical advice and fruitful discussions as well as Anni Nagy for support with illustrations.

## Author contributions
A.S.W. and M.E.S. designed the study. A.S.W. and A.B. carried out the experiments. A.S.W., U.B., B.B., and B.O. performed data analysis and developed computer and machine learning algorithm tools for data evaluation. S.M., H.K., and F.H. developed and provided technical tools. B.V.I. performed pump implantations. A.S.W, B.O., and M.E.S. prepared figures and wrote the manuscript.

## Additional information

**Competing interests:** The University of Zurich holds joint patents with Novartis Pharma for antibodies against Nogo-A and their use in neurological diseases. M.E.S is a board member of the spin-off company NovaGo Therapeutics Inc. The remaining authors declare no competing financial interests.

