## [Peer review file · Nature Communications]

Reviewers' comments:

Reviewer #1 (Remarks to the Author):

In this manuscript, Wahl et al report their results showing that optogenetic stimulation of the intact corticospinal tract (CST) axons after unilateral stroke is able to stimulate sprouting and results in functional recovery in relevant behavioral tasks. Together with a rehabilitation training, such opto-stimulation led to almost full recovery in the food pellet retrieval assay. Such analyses were performed by an automatic analysis of paw posture, suggesting that such functional recovery is due to the restoration instead of compensation. These results should be important for the stroke field. However, the anatomical basis for such nearly full functional recovery should be further examined. For example, how comparable are the distribution patterns of sprouting CST axons in the OptoStim group vs. intact ones? Such information should provide important insights into the structural requirements of functional recovery.

Other comments:

Fig. 1: In 1F, the authors claimed that about 40% of layer V cortical neurons were expressing mCherry throughout the sensorimotor cortex. Are these mCherry+ corticospinal neurons evenly distributed in premotor and somatosensory cortex? What is the explanation for the results that sprouting of such a subset of CST axons could restore full function in the denervated spinal cord. This concern also applies to Fig. 7A-B.

Fig. 3: It is technically commended to introduce an unbiased, automatic paw trajectory tracking and analysis method. However, the authors should compare the results from this and other available methods and point out what new information could be obtained with this method.

Fig. 4: In the Opto group (blue) in which AAV-Cre was injected only to C5-C6, what is the explanation for enhanced sprouting seen at all cervical levels (from C3 to T1)?

Reviewer #2 (Remarks to the Author):

This study used optogenetics to activate corticospinal circuitry following a PT stroke which destroyed >95% of sensorimotor cortex, and observed complete restoration of motor patterns.

Contralesional corticospinal tract (CST) sprouting which terminates in the denervated spinal cord following stroke has been described in different therapeutic interventions, and a positive correlation between this sprouting and the motor recovery has been demonstrated. In this study, optogenetic activation of intact CST (contralesional) resulted in motor recovery following large sensorimotor stroke in rats. Completely automated analysis of forepaw grasping behavior before and after stroke demonstrated that CST activation leads to full recovery of forelimb function, as measured by single pellet grasping, narrow beam crossing and the horizontal ladder test. In addition, training following stroke has a synergistic effect with the optogenetic stimulation and caused enhanced recovery. BDA labelling showed that optogenetic stimulation causes a significantly increased number of fibers which cross midline in the spinal cord to innervate the ipsilesional cord at C3-C8. This suggests that the stimulation and/or training are sufficient to induce axonal growth into the denervated cord. This, and the motor recovery data, are reported to be similar to an Anti-Nogo A immunotherapy which the lab has pioneered before. Inhibition of CST fibers was studied, using Cre-dependent AAV with an inhibitory light driven proton pump (Arch T), in 3 regions: pre/rostral M2, forelimb motor cortex and primary sensory cortex, by implanting optical fibers to each region. This allows for optical silencing at each location independently. Optical silencing of midline crossing CST fibers originating from the forelimb motor cortex resulted in a drop of performance for all cortical areas silenced, but was only significant for forelimb motor

cortex. Important to note, the animals tested here were treated with Anti-Nogo A immunotherapy, as optical stimulation as a therapy was not technically possible (as the inhibition is also optical). Additional analysis shows that inhibition of the forelimb cortex resulted in too short grasping behavior, while silencing premotor induced too far grasping behavior in relation to the pellet.

The studies in the present manuscript show that selective corticospinal tract stimulation, which produces axonal fibers which cross the midline from the intact to the denervated spinal cord, is sufficient for a full motor recovery after a large stroke. As stated in the discussion, stimulation of corticospinal neurons may be a possible therapeutic target in human patients, but many more detailed studies will be needed in this regard. It is also worth noting that this kind of approach in humans would require viral gene delivery, long-distance and circuit-specific gene expression and brain surgery with implantation of an optical stimulator. This is not the simplest application to a human, and would be a daunting translational path for regulatory approval.

The automated kinematic analysis is a novel and promising method for studying motor recovery, specifically to begin differentiating between restoration of movement versus compensatory movements. The selective inhibition of brain areas, using retrograde ArchT vectors in the re-wired corticospinal neurons, and its effects on very fine motor skills is another promising application of optogenetics, and makes it possible to dissect rewired circuits and uncover the functions which they take over.

Major Criticisms

- 1) It is understandable that Anti-Nogo/Training was used in place of the OptoStim/Training group, as it was not technically possible to optically inhibit animals that already had stimulatory rhodopsins. Would it have been possible to use a different method of inhibition, such as Gi DREADDs, to be able to test the OptoStim group and thus produce a more coherent argument for the bidirectional modulation of recovery with optical stimulation? Instead of using light to drive the inhibition, cannulae with CNO could instead be used to achieve the same number of targets and independently of one another.
- 2) The authors presumably separate optical stimulation and behavioral training based on their previous result with anti-NOGO therapy. In this manuscript, it is important to state the rationale for not combining them.
- 3) Fig. 7. It would be a good addition to the consideration of mechanism for this optogenetic stimulation if a correlation analysis between midline crossing and behavioral performance, in addition to the current analysis focusing just on the high-performers, is reported.
- 4) To prove that baseline motor trajectories were recovered (as opposed to compensatory responses) this reviewer would want some assessment of how different the data would have looked for a compensatory response. Is the similarity in trajectories before/after treatment biological or due to lack of sensitivity of tracking system and quantification? (note that differences in the spontaneous recovery group are not valid for this question because they are confounded by poor performance).
- 5) In a supplement figure it would be helpful to see what the stimulation optrode array looks like, so that the reader can get a sense of the volume of cortex that has this array implanted into it.
- 6) Is there a control group for stroke+optrode implantation—with no stimulation? The optrode array implantation into the contralateral cortex to the stroke is likely to induce local lesions and neuroinflammation.
- 7) The computer analysis of reach is a sophisticated addition to the field and provides excellent comparative data across these experimental conditions. Because this approach is so novel, there is

not a background of published information with which to evaluate the changes observed after stroke. One way in which the reader could understand these variables in skilled reach in the rat, and how they change over time, is for the authors to published how these variables change during the initial learning of the skilled reach task (prior to the stroke in these studies).

Minor Criticisms

- 1) First paragraph of page 7. In parenthesis, describing Fig2A success at 4 weeks, two numbers are given before the "OptoStim/Training group"- if one of them is for the "Anti-Nogo" group, please label it as so.
- 2) The last paragraph of the discussion makes an analogy of the brain after stroke to the developing nervous system. The writers claim that the 2 phases of a developing nervous system, 1) formation of imprecise and redundant circuits and 2) use dependent selective stabilization, may also relate to stroke and its recovery. They then make the claim that their bouts of targeted stimulation may enhance the first of these phases and that behavioral rehabilitation may enhance the second. I find this analogy somewhat confusing since the majority of the enhancement is seen in the optical stimulation group, with or without training. Also, this analogy seems to imply a synergistic effect of the two therapies when it may only be additive. Further, the analogy would require a phase of exuberant axonal sprouting and then pruning. This has not been demonstrated in this model.
- 3) Use of "hand" in the text
- 4) Fig 4 does not indicate that the muscle names indicate below the x axis and how they were positioned. Presumably they are motor neuron pools for each of these muscles.

Reviewer #3 (Remarks to the Author):

This manuscript by the lab Schwab describes optogenetic cortical-spinal tract stimulation's effect on promoting recovery from a large stroke to the sensorimotor cortex. This work done in rat model of very high quality extends work by Wahl et al. published in science. Notably the work compares anti-Nogo pharmacological treatment to the brain stimulation optogenetic approach. In the end both treatments converge at nearly full recovery of forelimb function defined by pellet reaching and quantified using a machine vision approach. Importantly the authors also look at transfer of every two other motor tasks including narrow beam walking. A well done study relatively minor comments only (with the exception of #4)!!

Critique:

- 1 The authors report the stroke they create is a relatively large affecting the contralateral hemisphere and implying a sprouting model similar to what they've previously shown for anti-nogo. In the manuscript they report stroke sizes but only one histological image is shown in Figure 1. The shows within anterior medial motor cortex alone does not support the claim that these are particularly large strokes that also extend to sensorimotor areas. More complete lesion size analysis would be appropriate with additional example coronal sections showing the full extent of the lesion. Furthermore, it would be nice to see the location of these lesions superimposed onto a brain Atlas cartoon in a flattened type presentation.
- 2 The legend appears at the end of the figure making it somewhat hard to find which color plots are which.
- 3 Although the analysis and presentation of reaching data is of high quality, visual images of the paws in figure 3 are of lower quality making it difficult to assess the various types of grasps. It would help to outline the paws or show some segmentation of these images.

4 The motor mapping data, while I appreciated this direction in its current form I think it's very difficult to conclude a change in ipsilateral motor maps with optogenetic training. The maps show clustered active points but no statistics. The cluster in M1 is likely not significantly different from the distribution in the untrained animals. I would suggest removing the motor mapping or perhaps indicating the difference between contralateral and ipsilateral motor pathways. A bilateral motor map would be very interesting to support claims that the lesion is complete. The most interesting comparison would be the extent of uncrossed map in an unlesioned animal compared to the lesioned animal and the opto-training. Since the number of animals and stimulation points are relatively small the best option may be to remove this figure which is probably the least convincing aspect of this great manuscript.

5 A caveat needs to be added that the author's manipulations and circuit investigations are mostly restricted to the unlesioned spared hemisphere, this could ideally come in the introduction and could be part of discussing stroke size.

6 The paper contains a surprising number of typos and it also contains language that seems non-scientific "sleep mix" for anesthesia.

7 The computer code for paw tracking looks exceptional (available by request) I think there will be a larger chance of uptake by the community if this software listed on a website for downloading directly.

8 It would be nice to have an overall model for recovery maybe something like Fig 5B

Reviewer #4 (Remarks to the Author):

This is another impressive contribution from a group that has been at the forefront of advancing mechanistic understanding of recovery and identifying neural treatment targets for stroke and other CNS injuries. That cortical electrical stimulation of the contralesional hemisphere can promote its reinnervation of spinal cord has previously been established in rodent stroke models, and that its behavioral effects depend on skill training could be predicted based on this group's previous findings with anti-Nogo-A treatments. However, the functional relevance of the contralesional contribution has remained a point of uncertainty, and this uncertainty helps fuel ongoing controversies surrounding the role of the contralesional hemisphere in stroke recovery. In this context, this article's exceedingly compelling demonstration that the contribution of the contralesional cortex can be driven by stimulation+training to support "true" recovery of more normal movement patterns is of major importance. It has been extraordinarily challenging in rodent stroke models to clearly infer recovery. The authors tackled this problem by developing a new computer vision based analysis approach for analyzing patterns of forepaw grasping movements. In addition to savvy and rigorous use of cutting-edge viral approaches for optogenetic manipulations, this computer vision based analysis approach is extremely innovative. The article is likely to be highly cited article not only because it clarifies the potential for contra cortex to be driven to support post stroke motor recovery, it has also tackled a longstanding methodological obstacle to more informative preclinical studies.

Minor Issues

1. Because it is reasonable to suspect that contralesional reinnervation patterns vary with lesion extent, clearer presentation of this extent seems called for. There is a single coronal section in Fig. 2F (which labels as M1 what may be premotor, medial agranular, territory). The cartoon of the lesions in Fig. 5 shows the lesion extent extending fully to midline, which does not seem to match

the extent in Fig. 2F. A representative coronal series of sections with lesions would be preferable.

2. p. 7 (near bottom): It is not safe to assume that the reduction in grasping frequency is due to reduced motivation. For example, impairments in shoulder and elbow movements could reduce the frequency in which the paw is extended through the slot to perform the grasp. Even after the forepaw is extended it is very plausible that proximal and/or distal motor impairments impede how quickly grasps can be repeated. It is suggested that the authors either reword this to avoid implying that the effects are motivational, or provide data that supports that motivation is the underlying cause.

3. The distinct effects of optogenetic silencing of position 1 (premotor/rostral M1) versus position 2 (more central M1) are very intriguing because of their potential implications, but the interpretations of these effects are somewhat meek. Why are the opposite effects of position 1 and 2 inhibition not more fully canceled out by inhibiting all sites at once? Does the greater resemblance to effects of position 1 inhibition imply its greater contribution to functionally relevant reinnervation? These findings raise numerous additional questions that might warrant at least brief discussion if room could be made for it. For example, if only M1 proper had contributed to reinnervation, would the rats perpetually reach too far? Are there any known functional properties of these two regions that would predict this pattern of results?

4. There are overstatements in the interpretations of the behavioral results in the discussion. E.g., p. 14.: "The result shows a full and true restoration of forelimb function...." The results show a full restoration of a very important aspect of fine manual function. It is impossible to rule out that other forelimb movement abnormalities persisted, short of measuring all forelimb movements. This just calls for more specific wording.

5. Many of the grasping pictures are extremely dark such that it is impossible to make out the details of the paw shape, especially in Fig. 3. It may be that these pictures are intended to be representative of what the computer algorithm analyzed, but they need to be improved for purpose of reader comprehension. The authors might also consider improving the brightness/contrast of the supplemental movie.

VERY Minor:

1. This sentence in abstract is confusing:

"In recovered animals, optogenetic silencing of cortical sub-regions showed that re-wired, ipsilaterally projecting corticospinal neurons in premotor and primary motor area resulted in too long or too short targeting of the restored grasping function, thus identifying the reestablishment of specific and anatomically localized cortical microcircuits induced by successful rehabilitative strategies"

2. Second paragraph on Pg. 3: "This analysis should be automatic, non-invasive, and purely visual to avoid interference with the recovery process." This implies that non-visual measures necessarily influence recovery mechanisms, which is unlikely. The wording could also be taken to mean that the authors are referring to measures of visual (rather than motor) behaviors.

3. Figures and captions

-Fig. 2A: Vertical axis is missing a label

-The line plots in Fig. 6 are not explained in the caption.

- Fig. 7: The authors might consider indicating the the optical fiber implant positions in this figure (even though these are from the anti-Nogo cohort).

4. Methods on p. 30 ("Novel tasks"). Neither the beam nor ladder task can be considered to be good tests of grasping function. That they are novel tasks of forelimb motor function is sufficient for their purpose.

Reviewed by Theresa A. Jones

Reviewer #5 (Remarks to the Author):

In this manuscript, Wahl et al describe a new paradigm for post-stroke recovery based on optogenetic stimulation of corticospinal circuit after injury. The work is a conceptual follow-up of a 2014 study of the same group where something similar was shown using molecular therapy. Interestingly, this work also introduces a more qualitative analysis on the extent of recovery, employing a computer vision dissection of motion that uses unsupervised learning to classify grasp movements.

Overall, this is clearly a well-designed study with properly described experiments but not always properly described results. I can offer some minor comments on the presentation and discussion of results and two slightly major concerns on the computational and statistical approach.

Figures 1 and 2.

The experimental setup is well explained. It was not clear from the main text whether all animals underwent AAV injection for Crimson expression but it seems this is the case judging from figure 1 and from the description of experimental procedures. The effect shown in figure 1 is clear and the statistics appropriate. Use of bar plots should, however, be discouraged as it is uninformative. Beeswarks and boxplots should be preferred.

Figure 3.

The manuscript heavily relies on the use of a new software. The algorithmic description of the procedure is technically well done but it is way too technical for the expected readership of the work; also, I find somehow unacceptable that software would be available only upon request. Public repositories of software do exist and it should be fairly easy for the authors to upload it and make it publicly available to readers and reviewers.

From the experimental point of view, the analysis shown in figure 3 seems to lack a control. Grasping analysis comparing BL to 35d in sham treated animal would suffice and would provide enough Ns to make a stronger case on the discerning abilities of this approach.

Finally, all the diagrams in Figure 3 seem to be poorly labelled/annotated. The significance of colours and lines could be better explained; the Ns of grasping event could be highlighted; axis on figures could be better labelled avoiding unnecessary abbreviations and using units when possible.

Figure 4,5

Nothing to say on the experimental aspect but data, again, could be presented in a better form. For instance: again, avoid bar graphs in figure 4b and label X axis. Properly label y axis on figure 4c (that's density); and properly rearrange x axis.

Figure 6 is really not clear. It is not clear what the graphs in panels b,c,d represent. When is location probability calculated? What is the x axis on the graphs on the left side? is it time?

Figure 7. If I understand correctly, the correlative claim between recovery and histology is based on Ns ~3, 4. I am not convinced these numbers offer enough power to draw this conclusion and I doubt they would survive a more appropriate statistical comparison.

Recommendations:

1. please provide the software used for the CV analysis, with instructions on how to use it.
2. validate the UL algorithm for this use case comparing sham-treated animals
3. review all figures to improve presentation quality
4. either increase power for experiments shown in figure 7 or re-evaluate claims

Our detailed responses to the referees' comments are as follows:

Reviewer #1:

In this manuscript, Wahl et al report their results showing that optogenetic stimulation of the intact corticospinal tract (CST) axons after unilateral stroke is able to stimulate sprouting and results in functional recovery in relevant behavioral tasks. Together with a rehabilitation training, such opto-stimulation led to almost full recovery in the food pellet retrieval assay. Such analyses were performed by an automatic analysis of paw posture, suggesting that such functional recovery is due to the restoration instead of compensation. These results should be important for the stroke field. However, the anatomical basis for such nearly full functional recovery should be further examined. For example, how comparable are the distribution patterns of sprouting CST axons in the OptoStim group vs. intact ones? Such information should provide important insights into the structural requirements of functional recovery.

We have performed new experiments including anterograde tracings with BDA in intact animals and added the analysis of ipsilaterally projecting and midline-crossing CST fibers into the cervical spinal cord to our study (new Fig. 4, A-F). Density of sprouting BDA labeled CST fibers in the denervated spinal cord was up to 16-18 times higher, in particular in lamina 6/7 of the ventral horn, in the 'OptoStim/Training' and 'Anti-Nogo/Training' group compared to BDA positive fibers in intact ('Naïve') animals (Fig. 4F). In addition, animals in the 'OptoStim' group revealed a 9-13x higher BDA positive CST fibers density e.g. in lamina 4/5 compared to intact animals (page 10 of the revised manuscript).

Other comments:

Fig. 1: In 1F, the authors claimed that about 40% of layer V cortical neurons were expressing mCherry throughout the sensorimotor cortex. Are these mCherry+ corticospinal neurons evenly distributed in premotor and somatosensory cortex?

We extended our analysis of mCherry/ChR2 expressing layer V cortical neurons to the premotor and somatosensory cortex: We find 39% of layer V cortical neurons expressing mCherry in the primary motor cortex which was significantly more than in the premotor (20%) and primary sensory cortex (5-7%) as presented now in Fig. 1F.

What is the explanation for the results that sprouting of such a subset of CST axons could restore full function in the denervated spinal cord. This concern also applies to Fig. 7A-B.

We have expanded our anatomical analysis and did not only examine the effect of the different rehabilitative treatments on CST fiber sprouting crossing from the intact cervical hemi cord to the denervated one (Fig. 4A-C), but also on sprouting of ipsilaterally and ventrally projecting, pre-existing CST fibers. In addition to enhanced "midline" crossing CST fibers sprouting, we also find pronounced sprouting of ipsilateral CST fibers in animals in the 'OptoStim/Training' and 'Anti-Nogo/Training' group which revealed full recovery of skilled motor function of the impaired paw (new Fig. 4D). Increased activity to the CST by optogenetics may have also stimulated corticospinal neurons to sprout into other brain regions such as the red nucleus and other brain stem areas (reticular formation, raphe nucleus) as previously described (Mosberger et al., 2017; Bachmann et al., 2014; Zörner et al., 2014). We have discussed this point in the new discussion section (p. 14).

Fig. 3: It is technically commended to introduce an unbiased, automatic paw trajectory tracking and analysis method. However, the authors should compare the results from this and other available methods and point out what new information could be obtained with this method.

Conventional methods include manual analysis such as the 10 point evaluation suggest by Whishaw et al., 2008, which is time consuming and often subjective, or kinematic analysis of grasping trajectories (Azim et al., 2014). However, pure grasping trajectories are not sensitive enough to detect delicate disparity of distinct aspects of the grasping act such as supination, targeting or paw closure (Wahl et al., 2014). In contrast, our approach has several advantages: Our algorithm allows an automatic and unsupervised analysis of grasping kinematics, without tattoos or other visual markers and no high speed recording. Furthermore, our method combines a single frame analysis with a sequence matching approach, meaning that each recorded image of a grasping sequence is ranked for its timely aspect of appearance and detailed paw shape analysis. Our algorithm thus allows the classification of each grasp closer or further away from the previously identified typical healthy grasping sequence. We have included this comparison now in the discussion section of the manuscript (p. 15).

Fig. 4: In the Opto group (blue) in which AAV-Cre was injected only to C5-C6, what is the explanation for enhanced sprouting seen at all cervical levels (from C3 to T1)?

Although the AAV-Cre virus was just injected to spinal cord level C5-C6, we find a large proportion of layer V neurons in M1 (~39%), M2 (~20%) and S1(~7%) expressing Channelrhodospin 2 in the end (Fig. 1F). Increasing activity of these neurons using optogenetic activation may have not only enhanced overall cervical sprouting but also sprouting in other brain(stem) regions as discussed above and in the discussion section (p. 14). Furthermore for the OptoStim group (blue group) we found only on level C4-C6 a significantly pronounced sprouting of midline crossing fibers compared to the 'Spontaneous Recovery', 'Delayed Training' and 'Naïve' animals (Fig. 4C). Sprouting of pre-existing ipsilaterally projecting ventral CST fibers was only significantly enhanced in the 'OptoStim' animals compared to the 'naïve' situation (Fig. 4D).

Reviewer #2 (Remarks to the Author):

This study used optogenetics to activate corticospinal circuitry following a PT stroke which destroyed >95% of sensorimotor cortex, and observed complete restoration of motor patterns.

Contralesional corticospinal tract (CST) sprouting which terminates in the denervated spinal cord following stroke has been described in different therapeutic interventions, and a positive correlation between this sprouting and the motor recovery has been demonstrated. In this study, optogenetic activation of intact CST (contralesional) resulted in motor recovery following large sensorimotor stroke in rats. Completely automated analysis of forepaw grasping behavior before and after stroke demonstrated that CST activation leads to full recovery of forelimb function, as measured by single pellet grasping, narrow beam crossing and the horizontal ladder test. In addition, training following stroke has a synergistic effect with the optogenetic stimulation and caused enhanced recovery. BDA labelling showed that optogenetic stimulation causes a significantly increased number of fibers which cross midline in the spinal cord to innervate the ipsilesional cord at C3-C8. This suggests that the stimulation and/or training are sufficient to induce axonal growth into the denervated cord. This, and the motor recovery data, are reported to be similar to an Anti-Nogo A immunotherapy which the lab has pioneered before. Inhibition of CST fibers was studied, using Cre-dependent AAV with an inhibitory light driven proton pump (Arch T), in 3 regions: pre/rostral M2, forelimb motor cortex and

primary sensory cortex, by implanting optical fibers to each region. This allows for optical silencing at each location independently. Optical silencing of midline crossing CST fibers originating from the forelimb motor cortex resulted in a drop of performance for all cortical areas silenced, but was only significant for forelimb motor cortex. Important to note, the animals tested here were treated with Anti-Nogo A immunotherapy, as optical stimulation as a therapy was not technically possible (as the inhibition is also optical). Additional analysis shows that inhibition of the forelimb cortex resulted in too short grasping behavior, while silencing premotor induced too far grasping behavior in relation to the pellet.

The studies in the present manuscript show that selective corticospinal tract stimulation, which produces axonal fibers which cross the midline from the intact to the denervated spinal cord, is sufficient for a full motor recovery after a large stroke. As stated in the discussion, stimulation of corticospinal neurons may be a possible therapeutic target in human patients, but many more detailed studies will be needed in this regard. It is also worth noting that this kind of approach in humans would require viral gene delivery, long-distance and circuit-specific gene expression and brain surgery with implantation of an optical stimulator. This is not the simplest application to a human, and would be a daunting translational path for regulatory approval.

The automated kinematic analysis is a novel and promising method for studying motor recovery, specifically to begin differentiating between restoration of movement versus compensatory movements. The selective inhibition of brain areas, using retrograde ArchT vectors in the re-wired corticospinal neurons, and its effects on very fine motor skills is another promising application of optogenetics, and makes it possible to dissect rewired circuits and uncover the functions which they take over.

Major Criticisms

1) It is understandable that Anti-Nogo/Training was used in place of the OptoStim/Training group, as it was not technically possible to optically inhibit animals that already had stimulatory rhodopsins. Would it have been possible to use a different method of inhibition, such as Gi DREADDs, to be able to test the OptoStim group and thus produce a more coherent argument for the bidirectional modulation of recovery with optical stimulation? Instead of using light to drive the inhibition, cannulae with CNO could instead be used to achieve the same number of targets and independently of one another.

We have previously shown that silencing newly sprouted CST fibers in animals which have received Anti-Nogo immunotherapy followed by intensive rehabilitative training after stroke using Gi DREADD technology results in a significant decline of regained grasping function (Wahl et al., 2014). However, our aim of the here presented study was to be able to specifically inhibit distinct contralesional brain sub-regions sending corticospinal projections and thus be able to identify region-specific reorganization responsible of specific aspects of regained grasping function, which was only possible with a Cre-dependent expression of Channelrhodopsin and several optical fiber implantations. Furthermore, there would have been serious technical caveats to a combined optogenetic and Gi DREADD approach: Before stroke we injected the AAV-Cre virus in the future contralesional cervical hemi-spinal cord, followed by an injection of the Cre-dependant Channelrhodopsin virus in the future contralesional hemisphere, where we then chronically implanted the 3 optical implants for optogenetic stimulation over the craniotomy. Silencing the newly outsprouting corticospinal fibers with GiDREADD technology would have practically meant not only to inject for the second time the AAV-Cre virus at the same cervical spinal cord level as before stroke and thus causing increased spinal cord tissue damage. But worse than this we would have needed to remove the chronically implanted optical

fibers and inject the Gi DREADD virus causing even worse cerebral damage making a reliable further experiment with silencing distinct CST neurons in the behaving (grasping) animal impossible.

2) The authors presumably separate optical stimulation and behavioral training based on their previous result with anti-NOGO therapy. In this manuscript, it is important to state the rationale for not combining them.

We have revised the formulations and now state (page 5): ‘We here used the sequential approach of first optogenetic stimulation of the intact corticospinal tract followed by intensive training based on our previous experience that early combination of two plasticity stimulating approaches could be detrimental [22].’

3) Fig. 7. It would be a good addition to the consideration of mechanism for this optogenetic stimulation if a correlation analysis between midline crossing and behavioral performance, in addition to the current analysis focusing just on the high-performers, is reported.

This is a great suggestion and we included a correlation analysis between midline crossing CST fibers and behavioral performance (new Fig. 4E). We found a direct correlation between the amount of midline crossing CST fibers and the level of skilled motor recovery.

4) To prove that baseline motor trajectories were recovered (as opposed to compensatory responses) this reviewer would want some assessment of how different the data would have looked for a compensatory response. Is the similarity in trajectories before/after treatment biological or due to lack of sensitivity of tracking system and quantification? (note that differences in the spontaneous recovery group are not valid for this question because they are confounded by poor performance).

We have previously shown (Wahl et al, 2014 – Supplementary Material), that pure trajectory analysis is insufficient to detect delicate differences of grasping quality. Our new computer vision based approach classifies not only single paw postures but also the correct sequence of paw postures and deviation in the impaired condition. The analysis of grasping sequences revealed e.g. for the ‘Spontaneous recovery’ group that animals typically reach too short (Fig. 3A), while in the ‘Delayed Training’ group the terminal supination movement before the close of the paw is limited (as written in the results section of the manuscript p.8/9). To determine the sensitivity of our approach we have used the algorithm to detect differences of grasping quality during the course of learning the single pellet grasping task (over 24 days before ‘baseline’ recordings prior to stroke). Our algorithm is able to classify even small differences in the healthy condition during the learning of the single pellet grasping. This is now described and documented in Supplementary Fig. 3 and the methods section.

5) In a supplement figure it would be helpful to see what the stimulation optrode array looks like, so that the reader can get a sense of the volume of cortex that has this array implanted into it.

We included such a supplementary figure (new ‘Supplementary Figure 2) showing the optrode array and depicting the estimated volume of the cortex exposed to the stimulation.

6) Is there a control group for stroke+optrode implantation—with no stimulation? The optrode array implantation into the contralateral cortex to the stroke is likely to induce local lesions and neuroinflammation.

All animals included in the study (Fig. 1) not only received spinal and cortical injections for the Cre-dependant expression of Channelrhodopsin but also implantation of the three optical fibers prior to stroke. Thus, after stroke animals could be randomized into the four different rehabilitation groups depending on their stroke lesion deficit in the single pellet grasping task. We experienced no infection or other signs of inflammation which would have required to remove animals from the study. As all animals underwent the same surgical intervention, differences in outcome cannot be explained by different severity of local lesions and neuroinflammation.

7) The computer analysis of reach is a sophisticated addition to the field and provides excellent comparative data across these experimental conditions. Because this approach is so novel, there is not a background of published information with which to evaluate the changes observed after stroke. One way in which the reader could understand these variables in skilled reach in the rat, and how they change over time, is for the authors to published how these variables change during the initial learning of the skilled reach task (prior to the stroke in these studies).

As the reviewer suggested, we analyzed additional recordings documented during the learning period before stroke to evaluate the ability of our automatic analysis to spot subtle changes in grasping behavior. Supplementary Figure 3 illustrates the evaluation graphically and highlights that even differences resulting from short-term learning can be measured. The p-values shown in Supplementary Figure 3D confirm these results numerically. The experiment demonstrates that our algorithm is sensitive enough to even detect small changes during the learning of single pellet grasping.

P-values between 0d and all time ranges:

	0d
From 2d until 8d	0.0474541785
From 14d until 17d	2.46E-014
From 18d until 24d	4.44E-018

Minor Criticisms

1) First paragraph of page 7. In parenthesis, describing Fig2A success at 4 weeks, two numbers are given before the “OptoStim/Training group” - if one of them is for the “Anti-Nogo” group, please label it as so.

Labelling of the ‘Anti-Nogo/Training’ group was included.

2) The last paragraph of the discussion makes an analogy of the brain after stroke to the developing nervous system. The writers claim that the 2 phases of a developing nervous system, 1) formation of imprecise and redundant circuits and 2) use dependent selective stabilization, may also relate to stroke and its recovery. They then make the claim that their bouts of targeted stimulation may enhance the first of these phases and that behavioral rehabilitation may enhance the second. I find this analogy somewhat confusing since the majority of the enhancement is seen in the optical stimulation group, with or without training. Also, this analogy seems to imply a synergistic effect of the two therapies

when it may only be additive. Further, the analogy would require a phase of exuberant axonal sprouting and then pruning. This has not been demonstrated in this model.

We have rewritten the last paragraph of the discussion and indicated the additive potential of a sequential combination of two therapies: ‘Both rehabilitation groups, the ‘Anti-Nogo/Training’ and the ‘OptoStim/Training’ combined an early, plasticity promoting treatment approach followed by intensive rehabilitative training of the impaired paw. This sequential combination resulted in almost full recovery of impaired motor function. In addition, for both treatment groups a similar sprouting pattern of CST fibers targeting motor neuronal pools in lamina 6/7 of the ventral horn in the denervated cervical spinal cord was found, suggesting at least an additive effect of the two therapies: First the stimulation of circuit formation and reorganization followed by use dependent selection, stabilization and strengthening of meaningful circuitry by rehabilitative training.’

3) Use of “hand” in the text#

‘Hand’ was replaced by ‘Paw’.

4) Fig 4 does not indicate that the muscle names indicate below the x axis and how they were positioned. Presumably they are motor neuron pools for each of these muscles.

We adjusted the figure caption for Fig. 4 to address this point: ‘Segment-specific analysis of midline crossing fibers showed that the two stimulation groups and the ‘Anti-Nogo/Training’ group had significantly more midline crossing fibers in the more rostral cervical spinal cord where motoneuron pools controlling proximal forelimb muscles are located, whereas fiber sprouting in the more caudal cervical spinal cord localizing motoneuron pools for the distal muscles are located was less pronounced compared to the ‘Spontaneous recovery’ and ‘Delayed training’ groups as well as in naïve animals.’

Reviewer #3 (Remarks to the Author):

This manuscript by the lab Schwab describes optogenetic cortical-spinal tract stimulation’s effect on promoting recovery from a large stroke to the sensorimotor cortex. This work done in rat model of very high quality extends work by Wahl et al. published in science. Notably the work compares anti-Nogo pharmacological treatment to the brain stimulation optogenetic approach. In the end both treatments converge at nearly full recovery of forelimb function defined by pellet reaching and quantified using a machine vision approach. Importantly the authors also look at transfer of every two other motor tasks including narrow beam walking. A well done study relatively minor comments only (with the exception of #4)!!

Critique:

1 The authors report the stroke they create is a relatively large affecting the contralateral hemisphere and implying a sprouting model similar to what they’ve previously shown for anti- nogo. In the manuscript they report stroke sizes but only one histological image is shown in Figure 1. The shows

within anterior medial motor cortex alone does not support the claim that these are particularly large strokes that also extend to sensorimotor areas. More complete lesion size analysis would be appropriate with additional example coronal sections showing the full extent of the lesion. Furthermore, it would be nice to see the location of these lesions superimposed onto a brain Atlas cartoon in a flattened type presentation.

We have added an extended histological stroke lesion analysis, including stroke lesion width, depth, length and location depending on bregma (new Supplementary Figure 1). In addition a series of coronar Nissl sections is shown where the position of primary and secondary motor and sensory cortices is indicated as determined by the Rat Brain Atlas by Paxinos and Watson 2005 (Supplementary Figure 1F).

2 The legend appears at the end of the figure making it somewhat hard to find which color plots are which.

We have added another legend at the center of the figure to increase readability of depicted plots.

3 Although the analysis and presentation of reaching data is of high quality, visual images of the paws in figure 3 are of lower quality making it difficult to assess the various types of grasps. It would help to outline the paws or show some segmentation of these images.

Better pictures are provided now; in Figure 2 and 3 we have increased contrast and intensity to increase visibility of paw shapes.

4 The motor mapping data, while I appreciated this direction in its current form I think it's very difficult to conclude a change in ipsilateral motor maps with optogenetic training. The maps show clustered active points but no statistics. The cluster in M1 is likely not significantly different from the distribution in the untrained animals. I would suggest removing the motor mapping or perhaps indicating the difference between contralateral and ipsilateral motor pathways. A bilateral motor map would be very interesting to support claims that the lesion is complete. The most interesting comparison would be the extent of uncrossed map in an unlesioned animal compared to the lesioned to animal and the opto-training. Since the number of animals and stimulation points are relatively small the best option may be to remove this figure which is probably the least convincing aspect of this great manuscript.

We included statistics for the intracranial microstimulation (ICMS) data of Fig. 7E: We find in position 2 – the primary motor cortex- a significantly higher EMG response in Anti-Nogo/Training animals compared to animals with spontaneous recovery, thus allowing at least a tentative conclusion that in animals with high levels of motor recovery not only distinct spinal but also cortical reorganization takes place. For the extent of ipsi- and contralateral EMG responses upon stimulation in the healthy contralesional hemisphere we discuss and refer to previous work from our group (Lindau et al., Brain, 2014).

5 A caveat needs to be added that the author's manipulations and circuit investigations are mostly restricted to the unlesioned spared hemisphere, this could ideally come in the introduction and could be part of discussing stroke size.

We included a sentence in the introduction stating our focus on the contralesional hemisphere: ‘. As we used a large sensorimotor stroke our study focused on the manipulation and circuit investigation of the contralesional hemisphere, as the main location of plastic remodeling and reorganizational

processes.'

6 The paper contains a surprising number of typos and it also contains language that seems non-scientific "sleep mix" for anesthesia.

'Sleep mix' was substituted by 'anesthesia'. We also carefully reviewed the manuscript for typos and corrected them.

7 The computer code for paw tracking looks exceptional (available by request) I think there will be a larger chance of uptake by the community if this software listed on a website for downloading directly.

We now provide the code and data together with a documentation describing the computational method of our behavior analysis as supplementary material. Due to data package size we were unable to upload the computer software to the online system of Nature communications.

However, the software is available on our server using the following link

<https://hcicloud.iwr.uni-heidelberg.de/index.php/s/VFhH4J0aupXqcbE>

The password to access the data is 'behavior'.

8 It would be nice to have an overall model for recovery maybe something like Fig 5B

We included an overall model in the new Fig. 7F.

Reviewer #4 (Remarks to the Author):

This is another impressive contribution from a group that has been at the forefront of advancing mechanistic understanding of recovery and identifying neural treatment targets for stroke and other CNS injuries. That cortical electrical stimulation of the contralesional hemisphere can promote its reinnervation of spinal cord has previously been established in rodent stroke models, and that its behavioral effects depend on skill training could be predicted based on this group's previous findings with anti-Nogo-A treatments. However, the functional relevance of the contralesional contribution has remained a point of uncertainty, and this uncertainty helps fuel ongoing controversies surrounding the role of the contralesional hemisphere in stroke recovery. In this context, this article's exceedingly compelling demonstration that the contribution of the contralesional cortex can be driven by stimulation+training to support "true" recovery of more normal movement patterns is of major importance. It has been extraordinarily challenging in rodent stroke models to clearly infer recovery. The authors tackled this problem by developing a new computer vision based analysis approach for analyzing patterns of forepaw grasping movements. In addition to savvy and rigorous use of cutting-edge viral approaches for optogenetic manipulations, this computer vision based analysis approach is extremely innovative. The article is likely to be highly cited article not only because it clarifies the potential for contra cortex to be driven to support post stroke motor recovery, it has also tackled a

longstanding methodological obstacle to more informative preclinical studies.

Minor Issues

1. Because it is reasonable to suspect that contralesional reinnervation patterns vary with lesion extent, clearer presentation of this extent seems called for. There is a single coronal section in Fig. 2F (which labels as M1 what may be premotor, medial agranular, territory). The cartoon of the lesions in Fig. 5 shows the lesion extent extending fully to midline, which does not seem to match the extent in Fig. 2F. A representative coronal series of sections with lesions would be preferable.

We have added an extended histological stroke lesion analysis, including stroke lesion width, depth, length and location depending on bregma (new Supplementary Figure 1). In addition a series of coronar Nissl sections is shown where the position of primary and secondary motor and sensory cortices is indicated as determined by the Rat Brain Atlas by Paxinos and Watson 2005 (Supplementary Figure 1F). We thank for the hint concerning the cartoons in Fig. 5 and re-adjusted their lesion sizes according to our results in Fig. 2 and Supplementary Figure 1.

2. p. 7 (near bottom): It is not safe to assume that the reduction in grasping frequency is due to reduced motivation. For example, impairments in shoulder and elbow movements could reduce the frequency in which the paw is extended through the slot to perform the grasp. Even after the forepaw is extended it is very plausible that proximal and/or distal motor impairments impede how quickly grasps can be repeated. It is suggested that the authors either reword this to avoid implying that the effects are motivational, or provide data that supports that motivation is the underlying cause.

We have re-written the paragraph and included the suggestions as follows: ‘Grasping frequency of all animals was reduced within the first week after stroke. Reduced grasping frequency can be due to decreased motivation or reduced grasping velocity because of proximal shoulder and elbow impairments. While grasping frequency quickly recovered to baseline levels within 2 weeks for all cohorts, the skilled behavior did not as investigated in Fig. 2C.’

3. The distinct effects of optogenetic silencing of position 1 (premotor/rostral M1) versus position 2 (more central M1) are very intriguing because of their potential implications, but the interpretations of these effects are somewhat meek. Why are the opposite effects of position 1 and 2 inhibition not more fully canceled out by inhibiting all sites at once?

For the well-recovered animals (‘Anti-Nogo/Training’ group) silencing at position 1 resulted in too long grasps (Fig. 6B), while silencing at position 2 caused too short grasping and thus missing the pellet. When silencing at all three positions at the same time, animals still show too long grasps but less significant compared to the silencing of the central M1 suggesting a more dominant effect of the Pos. 1, which is also statistically reflected (see level of significance of the KS-Test).

We have responded to this comment in the discussion section on page 16.

Does the greater resemblance to effects of position 1 inhibition imply its greater contribution to functionally relevant reinnervation? These findings raise numerous additional questions that might warrant at least brief discussion if room could be made for it. For example, if only M1 proper had contributed to reinnervation, would the rats perpetually reach too far? Are there any known functional properties of these two regions that would predict this pattern of results?

To address these questions we have extended the discussion section on page 17 in the following way:

‘Cortical motor maps as the basis of voluntary movements have been described extensively: In particular electrophysiological studies in monkeys, cats and rodents have detected reproducible, complex movements of the forelimbs and hindlimbs triggered from distinct positions of the motor cortex [47-49]: Harrison et al., e.g. demonstrated that the forelimb motor cortex is subdivided in functional subregions for abduction and adduction movements, while others report that motor cortex silencing in the intact rodent interferes with initiation or performance of trained fine motor tasks [50, 51]. In mice with forelimb motor cortex strokes, secondary induced strokes in the premotor cortex destroyed regained skilled grasping function suggesting an important role of the premotor cortex for the regain of lost forelimb function [52]. However, the ability of the contralesional pre- and motor cortex to induce local circuitry of distinct aspects for fine motor function after stroke has been not revealed before. We show here for the first time that after an injury such as stroke and a successful rehabilitative intervention, plastic, rewired neurons, - and presumably entire microcircuits, - are found in the corresponding anatomical positions and are involved in specific aspects of the grasping sequence.’

4. There are overstatements in the interpretations of the behavioral results in the discussion. E.g., p. 14.: "The result shows a full and true restoration of forelimb function...." The results show a full restoration of a very important aspect of fine manual function. It is impossible to rule out that other forelimb movement abnormalities persisted, short of measuring all forelimb movements. This just calls for more specific wording.

We re-wrote the paragraph as follows: ‘The kinematic analysis suggested a true recovery of the grasping behavior rather than compensatory movements.’

5. Many of the grasping pictures are extremely dark such that it is impossible to make out the details of the paw shape, especially in Fig. 3. It may be that these pictures are intended to be representative of what the computer algorithm analyzed, but they need to be improved for purpose of reader comprehension. The authors might also consider improving the brightness/contrast of the supplemental movie.

We have improved picture quality as well as increased brightness/contrast for all pictures in Fig. 2 and 3 as well as the supplementary movie.

VERY Minor:

1. This sentence in abstract is confusing:

"In recovered animals, optogenetic silencing of cortical sub-regions showed that re-wired, ipsilaterally projecting corticospinal neurons in premotor and primary motor area resulted in too long or too short targeting of the restored grasping function, thus identifying the reestablishment of specific and anatomically localized cortical microcircuits induced by successful rehabilitative strategies"

We thank the reviewer for this hint and have re-written the sentence as follows:

‘In recovered animals, optogenetic silencing of corticospinal projecting neurons in premotor and primary motor area resulted in too long or too short targeting of the restored grasping function, thus identifying the reestablishment of specific and anatomically localized cortical microcircuits induced by successful rehabilitative strategies.’

2. Second paragraph on Pg. 3: "This analysis should be automatic, non-invasive, and purely visual to avoid interference with the recovery process." This implies that non-visual measures necessarily influence recovery mechanisms, which is unlikely. The wording could also be taken to mean that the authors are referring to measures of visual (rather than motor) behaviors.

We have re-written the sentence to avoid misunderstandings as follows: 'This analysis should be automatic and non-invasive to avoid interference with the recovery process.'

3. Figures and captions

-Fig. 2A: Vertical axis is missing a label

A vertical axis legend was included.

-The line plots in Fig. 6 are not explained in the caption.

We revised the caption for Fig. 6 and included an explanation for the line plots.

- Fig. 7: The authors might consider indicating the the optical fiber implant positions in this figure (even though these are from the anti-Nogo cohort).

The positions of the optical fiber implants were included in the heat maps of the ICMS data in Fig. 7E.

4. Methods on p. 30 ("Novel tasks"). Neither the beam nor ladder task can be considered to be good tests of grasping function. That they are novel tasks of forelimb motor function is sufficient for their purpose.

We agree that beam and ladder are very insensitive tasks to assess skilled forelimb function. In the current study we used these tasks only as novel tasks to be able to assess non-task specific recovery of forelimb function.

Reviewer #5 (Remarks to the Author):

In this manuscript, Wahl et al describe a new paradigm for post-stroke recovery based on optogenetic stimulation of corticospinal circuit after injury. The work is a conceptual follow-up of a 2014 study of the same group where something similar was shown using molecular therapy. Interestingly, this work also introduces a more qualitative analysis on the extent of recovery, employing a computer vision dissection of motion that uses unsupervised learning to classify grasp movements.

Overall, this is clearly a well-designed study with properly described experiments but not always properly described results. I can offer some minor comments on the presentation and discussion of results and two slightly major concerns on the computational and statistical approach.

Figures 1 and 2.

The experimental setup is well explained. It was not clear from the main text whether all animals

underwent AAV injection for Crimson expression but it seems this is the case judging from figure 1 and from the description of experimental procedures. The effect shown in figure 1 is clear and the statistics appropriate. Use of bar plots should, however, be discouraged as it is uninformative. Beeswarms and boxplots should be preferred.

We changed part of the first paragraph of the results section to also clearly state in the main text that all animals have received AAV injections for Channelrhodopsin as well as fiber implantation before stroke and were then randomized in the rehabilitation groups as also described in the experimental procedures. To address the reviewer's request we also first transformed all bar plots of Fig. 1 and 2 into box-whisker plots but realized that – because of the amount of data per graph- readability and visibility suffered for most of the plots. This is why we decided to provide individual data superimposed to the bar plots to indicate data distribution and add box-whisker plots as supplementary (new Supplementary Figure 5).

Figure 3.

The manuscript heavily relies on the use of a new software. The algorithmic description of the procedure is technically well done but it is way too technical for the expected readership of the work; also, I find somehow unacceptable that software would be available only upon request. Public repositories of software do exist and it should be fairly easy for the authors to upload it and make it publicly available to readers and reviewers.

We now provide the code and data together with a documentation describing the computational method of our behavior analysis as supplementary material. Due to data package size we were unable to upload the computer software to the online manuscript submitting system of Nature communications.

However, the software is available on our server using the following link

<https://hcicloud.iwr.uni-heidelberg.de/index.php/s/VFhH4J0aupXqcbE>

The password to access the data is 'behavior'.

Upon acceptance we would like to suggest uploading the software to the publicly available platform 'github'.

From the experimental point of view, the analysis shown in figure 3 seems to lack a control. Grasping analysis comparing BL to 35d in sham treated animal would suffice and would provide enough Ns to make a stronger case on the discerning abilities of this approach.

In Figure 3 grasping performance 35 days after stroke of each rehabilitation group is compared to the grasping performance of the same animals at baseline as an internal control for each rehabilitation: Our aim here was to assess the recovery level of each cohort which was only possible by directly comparing pre- and post-stroke grasping trajectories within the same animal and average then for the whole rehabilitation group. This is why we did not include sham-operated animals in Fig. 3. However we have now added the analysis of grasping trajectories of sham-operated animals, which is provided in Supplementary Figure 4. We did not find a difference in paw posture between baseline and 35 days after sham surgery as stated in the manuscript on page 9.

Finally, all the diagrams in Figure 3 seem to be poorly labelled/annotated. The significance of colours and lines could be better explained; the Ns of grasping event could be highlighted; axis on figures could be better labelled avoiding unnecessary abbreviations and using units when possible.

We fully reviewed Figure 3 and revised in particular the figure caption and included the Ns of grasping events.

Figure 4,5

Nothing to say on the experimental aspect but data, again, could be presented in a better form. For instance: again, avoid bar graphs in figure 4b and label X axis. Properly label y axis on figure 4c (that's density); and properly rearrange x axis.

As already described for Fig. 1 and 2 above: Because of the amount of data per graph readability and visibility suffered for most of the plots using box-whisker plots. Thus we included individual data superimposed to the bar plots to indicate data distribution. We also properly labeled all x and y axis and re-wrote figure captions.

Figure 6 is really not clear. It is not clear what the graphs in panels b,c,d represent. When is location probability calculated? What is the x axis on the graphs on the left side? is it time?

We fully revised the caption for Fig. 6 and also included the requested information as follows:

‘(B)-(D) Spatial distribution of the furthest extension calculated using every grasping trial under ‘light off’ (I) and ‘light on’ (II) conditions; (III) relative distance between ‘light off’ and ‘light on’ (*=position of the sugar pellet). Therefore, the spatial distribution (I) and (II) are marginalized onto the x-location, before subtracting the resulting 1-dimensional distributions to obtain (III).’

Figure 7. If I understand correctly, the correlative claim between recovery and histology is based on Ns ~3, 4. I am not convinced these numbers offer enough power to draw this conclusion and I doubt they would survive a more appropriate statistical comparison.

We included statistics for the intracranial microstimulation (ICMS) data of Fig. 7E: We find in position 2 – the primary motor cortex- a significantly higher EMG response in Anti-Nogo/Training animals compared to animals with spontaneous recovery, thus allowing at least a tentative conclusion that in animals with high levels of motor recovery not only distinct spinal but also cortical reorganization takes place. We also re-wrote our claims in the results section as follows:

‘Our results suggest not only a distinct sprouting pattern of CST fibers from the contralesional hemisphere into the ventral horn of the denervated cervical hemi-cord, but also provide hints for a regionalized functional reorganization in the contralesional pre- and primary motor cortex.’

Recommendations:

1. please provide the software used for the CV analysis, with instructions on how to use it.

As said above, the software is provided as supplementary material. If this is not possible due to the data package size we suggest uploading our software to the publicly available platform 'github'.

2. validate the UL algorithm for this use case comparing sham-treated animals

We included the behavior analysis of our sham-treated animals as an add-on to Figure 3 in Supplementary Figure 4 and described our result on page 9 of the manuscript.

3. review all figures to improve presentation quality

We reviewed all figures to improve readability and visibility according to all reviewers' comments and concerns.

4. either increase power for experiments shown in figure 7 or re-evaluate claims

As described above we included the statistics for Fig. 7E and found that at least for Pos. 2 there was a significantly enhanced EMG response for the 'Anti-Nogo/Training' group compared to the 'Spontaneous recovery' group, thus justifying the small sample size. However we still attenuated our claims and re-wrote the paragraph in the results section (page 13).

We thank all reviewers for the helpful comments and hope that we have now addressed all comments and concerns.

Reviewers' comments:

Reviewer #1 (Remarks to the Author):

This revised manuscript, with added new data, is significantly improved. However, there is a few minor issues to be addressed prior to publication:

1. The temporal specificity of optogenetic silencing.

As shown in Fig. 5D, once the regional optogenetic silencing starts, it covers the whole reaching and grasping process and even the unrelated behaviors during the 100 s. It will be more meaningful if the light turns on at a specific time point (as shown in Fig. 3, the authors divided the reaching and grasping by five time points) to see how (if any) the optogenetic silencing of the 3 sites affect the behavior.

2. The effects of optogenetic silencing on the reaching behavior.

In Fig. 6, the authors use machine learning algorithms to detect the differences between pre- and post-optogenetic silencing. They concluded that optogenetic inhibition of M1 and M2 caused "too short grasp" or "too far grasp", respectively. Such description is vague and without specific quantification. In addition, one would wonder why the authors didn't apply the same method as shown in Fig. 3 to compare paw trajectories pre- and post-optogenetic silencing.

3. The labeling efficiency of CSNs at M2, M1 and S1.

By injecting AAV-Cre at C5-C6, the CSNs are unevenly targeted within the cortex. As shown in Fig. 7B, the cell density is highest in M2, but much less in S1. Therefore, the results of optogenetic inhibition could be biased between different cortical areas. The same issue exists for Fig. 1F. The authors should discuss this issue.

Reviewer #2 (Remarks to the Author):

This revised manuscript meets the original concerns of this reviewer. The revision includes new tracing studies, no correlational statistical analyses, additional lesion analysis and modified text.

Reviewer #3 (Remarks to the Author):

For some reason I could only see the new supplemental figures which address my points (I could not see the new main figures).

The authors also outline their changes in the response to reviewers satisfying me.

This is a well done and extensive study I have nothing further.

Reviewer #4 (Remarks to the Author):

The manuscript by Wahl et al has been substantially revised. Prior issues with overstatements/assumptions in the interpretations have been address. Prior issues with figure quality have been nicely remedied. A new Supp. Fig. 1 now clarifies lesion characteristics well. Several other new analyses strengthen the inferences. This includes interesting new data on contributions of the ipsilateral CST pathway of contralesional M1 to reinnervation of denervated cervical cord. The re-submission maintains the strengths that I previously noted. Its compelling

demonstration that contralesional cortical contributions to reinnervation after stroke can be driven with stimulation+training to support "true" recovery and the innovative new computer vision based approach for assessing recovery are high impact contributions. The few remaining issues should be easy to address.

Issues:

The representative image in Fig 2F continues to be problematic. As noted previously, an "M1" label is approximately centered within the M2 anatomical subregion in this section. However, Supp. Fig 1 now makes it clear that this image is a very poor choice as a representative example. It would be far better to show a coronal section that is nearer the middle of the anterior to posterior extent of the lesion and that captures the near-total destruction of M1 that resulted from the infarcts. One of the coronal sections that is on either side of bregma in the new Supp. Fig.1 would do.

Why does the OptoStim/Training group lack data points on days 15 and 16 in Fig. 2A?

The significance of the new correlation between CST fibers and motor recovery that is shown in Fig. 4E should be reported.

The ladder and beam tasks continue to be referred to as "tasks of grasping function" on p. 31. Neither task is sensitive nor specific to grasping function, and thus this wording is misleading and inappropriate. It would be appropriate to refer to these tasks as measures of forelimb sensorimotor function.

An addition to the introduction to explain the rationale of the focus on contralesional cortex is problematic as worded: "As we used a large sensorimotor stroke our study focused on the manipulation and circuit investigation of the intact, contralesional hemisphere, as the main location of plastic remodeling and reorganizational processes." I think the authors would agree that major remodeling/reorganization in both hemispheres can be expected after such large sensorimotor cortical infarcts. They may mean that it is the main region in which remodeling/reorganization that mediates recovery of movement in the paretic side is likely to be found.

Minor

The authors might consider separately reporting the EMG responses for wrist, elbow and shoulder that were evoked from the M1 (Pos. 2) hotspot, for example, in a supplementary table,

The authors might inform the reader that the right side is the paretic side in the Supplementary movie.

The blue line that presumably represents the OptoStim group in Fig 2C is dissimilar to its defined color code.

There continues to be a need for more careful editing.

- Caption to Fig. 1, the abbreviation for primary motor cortex is defined as "M2"
- Caption to Supp Fig. 1: "coronar" should be "coronal", "secondary" should "secondary"
- Second to last paragraph of p. 10, "ilpsilaterally" should be "ipsilaterally" and "innovation" should be "innervation".
- New text on p. 16, "subjection" should be "subjective"

Reviewer #5 (Remarks to the Author):

I am satisfied with the changes the authors made to the manuscript. Figures are clearer and all claims are proportioned to the findings.

I think uploading the software to github is definitely the best option - as long as the github link appears in the accepted manuscript, of course.

Also, I strongly recommend embedding the control shown in Supplementary Figure 4 directly in Figure 3. There is no reason for it to be relegated to supplementary results.

Giorgio Gilestro
Imperial College London

Our detailed responses to the referees' comments are as follows:

Reviewer #1:

This revised manuscript, with added new data, is significantly improved. However, there is a few minor issues to be addressed prior to publication:

1. The temporal specificity of optogenetic silencing.

As shown in Fig. 5D, once the regional optogenetic silencing starts, it covers the whole reaching and grasping process and even the unrelated behaviors during the 100 s. It will be more meaningful if the light turns on at a specific time point (as shown in Fig. 3, the authors divided the reaching and grasping by five time points) to see how (if any) the optogenetic silencing of the 3 sites affect the behavior.

In our stimulation paradigm the initiation of the three lasers was regulated by a light barrier positioned at the centre of the provided food pellet so that by the first grasp through the light barrier a program would start which would keep the lasers off for 100s and turned them on afterwards for a 100s. The paradigm was developed to compare physiological grasping behaviour in a distinct time frame without and with selected inhibition of distinct brain regions rigorously instead of manipulating individual grasps, which would increase a random effect: Kinematics of individual grasps already tend to be dissimilar even in the healthy naïve situation. The reviewer's suggestion of testing the effect of different silencing paradigms and thus manipulate grasping behaviour at different time points would be definitely interesting. But it is beyond the scope of this paper and would represent a total new study of its own, as several reports in the literature have already demonstrated that different optogenetic stimulation/inhibition schemes result in a different behavioural phenotype (Chiu et al., 2014; Park et al., 2016; Roy et al., 2016). Several studies inhibited specific cell populations or even brain areas by the same continuous light stimulation as we did for up to even several minutes (Han et al., 2011; Miao et al., 2015; Tecuapetla et al., 2014), as the time off kinematics of the engineered proton pump ArchT is relatively slow compared to fast speed opsins such as Chronos (Chow et al., 2010; Mattis et al., 2011). A dissection of different grasping phases by ArchT stimulation would be difficult or impossible due to these slow kinetics of ArchT, in particular as the grasping velocity in trained rats is below 100ms. Optogenetically inhibiting individual phases of a grasp would therefore require a different, faster silencing construct, the testing of different stimulation paradigms, and a different protocol and experimental set-up for laser stimulation. This would clearly be beyond the scope of this paper.

2. The effects of optogenetic silencing on the reaching behavior.

In Fig. 6, the authors use machine learning algorithms to detect the differences between pre- and post-optogenetic silencing. They concluded that optogenetic inhibition of M1 and M2 caused "too short grasp" or "too far grasp", respectively. Such description is vague and without specific quantification. In addition, one would wonder why the authors didn't apply the same method as shown in Fig. 3 to compare paw trajectories pre- and post-optogenetic silencing.

We agree with the comments of the reviewer and have added additional data and analyses to the paper. In Supplementary Table 1 we have now additionally quantified the asserted differences of the grasping length between light-on and light-off sessions. For every cohort and position we calculated a weighted mean/std x-coordinate of commonly occurring coordinates (local maxima of Fig. 6 B(I)/B(II), C(I)/C(II) and D(I)/D(II) after projecting the 2D matrix to the x-axis) during light-off and

light-on sessions separately. Afterwards we subtracted the results of light-off and light-on phases and converted the values from pixel into mm. A positive mean indicates longer grasps during a light-on session. Quantitative measurement is now provided, which shows that animals of the Anti-Nogo/Training cohort grasp longer in Position 1 (on average $3.4\text{ mm}\pm 0.9$) and shorter in Position 2 ($-1.98\text{ mm}\pm 0.7$).

While the method developed and described in Fig. 3 examines multiple aspects of grasping trajectories, kinematics and paw postures, we suspected that the silencing of newly out-sprouting corticospinal fibers originating from the contralesional hemisphere and terminating on C5, C6 level of the cervical spinal cord primarily influences the targeting aspect of grasping due to some evidence gained from previous work (Wahl et al., 2014) using the less specific chemogenetic silencing.

3. The labeling efficiency of CSNs at M2, M1 and S1.

By injecting AAV-Cre at C5-C6, the CSNs are unevenly targeted within the cortex. As shown in Fig. 7B, the cell density is highest in M2, but much less in S1. Therefore, the results of optogenetic inhibition could be biased between different cortical areas. The same issue exists for Fig. 1F. The authors should discuss this issue.

For the stimulation of intact corticospinal fibers to promote functional recovery after stroke we labelled CST fibers terminating at C5/C6 spinal cord level originating from the healthy hemisphere with a Cre-dependant approach before the stroke (Fig. 1A). Optogenetic stimulation of the corticospinal projecting neurons expressing Channelrhodopsin 2 was sufficient to induce fiber growth in the denervated cervical spinal cord. In contrast, for the silencing approach viral vectors for the expression of the inhibitory light sensitive proton pump ArchT were injected after stroke and the rehabilitation phase (Fig. 5A) as we aimed to target specifically the reorganized and newly sprouted corticospinal fibers. Fig. 7C shows a significantly lower number of GFP positive neurons expressing ArchT in animals with spontaneous recovery than in animals of the Anti-Nogo/Training group. This is consistent with the data in Fig. 4, where the Anti-Nogo/Training group revealed a significantly enhanced corticospinal fiber sprouting compared to the ‘Spontaneous recovery’ group. We discussed this and added in the manuscript on page 17: ‘a significantly reduced cell density expressing ArchT had been detected in the ‘Spontaneous recovery’ group, which may furthermore contribute to the reduced effect of optogenetic silencing in those animals.’ Regarding to the uneven distribution of the ArchT cell density in the cortex of Anti-Nogo/ Training animals we added on page 17: “Another aspect may be the uneven distribution of rewired corticospinal projecting fibers with a higher cell density expressing ArchT in the pre-motor and motor cortex versus S1 (Fig. 7B) which may also have influenced the magnitude of the optogenetic silencing effect in the different positions (Fig. 5C) in the ‘Anti-Nogo/ Training’ group.”

References

- Chiu, W.T., Lin, C.M., Tsai, T.C., Wu, C.W., Tsai, C.L., Lin, S.H., and Chen, J.J. (2014). Real-time electrochemical recording of dopamine release under optogenetic stimulation. *PloS one* 9, e89293.
- Chow, B.Y., Han, X., Dobry, A.S., Qian, X., Chuong, A.S., Li, M., Henninger, M.A., Belfort, G.M., Lin, Y., Monahan, P.E., *et al.* (2010). High-performance genetically targetable optical neural silencing by light-driven proton pumps. *Nature* 463, 98-102.

Han, X., Chow, B.Y., Zhou, H., Klapoetke, N.C., Chuong, A., Rajimehr, R., Yang, A., Baratta, M.V., Winkle, J., Desimone, R., *et al.* (2011). A high-light sensitivity optical neural silencer: development and application to optogenetic control of non-human primate cortex. *Front Syst Neurosci* 5, 18.

Mattis, J., Tye, K.M., Ferenczi, E.A., Ramakrishnan, C., O'Shea, D.J., Prakash, R., Gunaydin, L.A., Hyun, M., Fenno, L.E., Gradinaru, V., *et al.* (2011). Principles for applying optogenetic tools derived from direct comparative analysis of microbial opsins. *Nature methods* 9, 159-172.

Miao, C., Cao, Q., Ito, H.T., Yamahachi, H., Witter, M.P., Moser, M.B., and Moser, E.I. (2015). Hippocampal Remapping after Partial Inactivation of the Medial Entorhinal Cortex. *Neuron* 88, 590-603.

Park, S., Bandi, A., Lee, C.R., and Margolis, D.J. (2016). Peripheral optogenetic stimulation induces whisker movement and sensory perception in head-fixed mice. *Elife* 5.

Roy, D.S., Arons, A., Mitchell, T.I., Pignatelli, M., Ryan, T.J., and Tonegawa, S. (2016). Memory retrieval by activating engram cells in mouse models of early Alzheimer's disease. *Nature* 531, 508-512.

Tecuapetla, F., Matias, S., Dugue, G.P., Mainen, Z.F., and Costa, R.M. (2014). Balanced activity in basal ganglia projection pathways is critical for contraversive movements. *Nature communications* 5, 4315.

We thank reviewer #2 and #3 for their helpful comments and are glad that our responses satisfied their concerns.

Reviewer #4:

The manuscript by Wahl et al has been substantially revised. Prior issues with overstatements/assumptions in the interpretations have been address. Prior issues with figure quality have been nicely remedied. A new Supp. Fig. 1 now clarifies lesion characteristics well. Several other new analyses strengthen the inferences. This includes interesting new data on contributions of the ipsilateral CST pathway of contralesional M1 to reinnervation of denervated cervical cord. The re-submission maintains the strengths that I previously noted. Its compelling demonstration that contralesional cortical contributions to reinnervation after stroke can be driven with stimulation+training to support "true" recovery and the innovative new computer vision based approach for assessing recovery are high impact contributions. The few remaining issues should be easy to address.

Issues:

The representative image in Fig 2F continues to be problematic. As noted previously, an "M1" label is approximately centered within the M2 anatomical subregion in this section. However, Supp. Fig 1 now makes it clear that this image is a very poor choice as a representative example. It would be far better to show a coronal section that is nearer the middle of the anterior to posterior extent of the lesion and that captures the near-total distruction of M1 that resulted from the infarcts. One of the coronal sections that is on either side of bregma in the new Supp. Fig.1 would do.

We thank this reviewer for this hint and changed the image in Fig. 2F revealing the near-total destruction of M1 as a result of the infarct.

Why does the OptoStim/Training group lack data points on days 15 and 16 in Fig. 2A?

The experiments described in Fig. 1 were performed first and the best group (the OptoStim/Training group) was then compared with animals which first received Anti-Nogo immunotherapy followed by intensive rehabilitative training (Fig. 2A). While we were conducting the rehabilitative training for Fig. 1 experiments, unannounced, noisy construction works happened in our animal facility on days 15 and 16. Due to this the data gained on days 15 and 16 after stroke were inconsistent (Fig 2A) and could thus not be included in the final analysis. We have mentioned this issue in the experimental set-up description on page 20 as one of the exclusion criteria.

The significance of the new correlation between CST fibers and motor recovery that is shown in Fig. 4E should be reported.

We thank the reviewer for this hint and included the test and significance level for Fig. 4E in the figure and caption.

The ladder and beam tasks continue to be referred to as "tasks of grasping function" on p. 31. Neither task is sensitive nor specific to grasping function, and thus this wording is misleading and inappropriate. It would be appropriate to refer to these tasks as measures of forelimb sensorimotor function.

The words "grasping function" on p. 31/new p. 27 were replaced by "forelimb sensorimotor function" as suggested.

An addition to the introduction to explain the rationale of the focus on contralesional cortex is problematic as worded: "As we used a large sensorimotor stroke our study focused on the manipulation and circuit investigation of the intact, contralesional hemisphere, as the main location of plastic remodeling and reorganizational processes." I think the authors would agree that major remodeling/reorganization in both hemispheres can be expected after such large sensorimotor cortical infarcts. They may mean that it is the main region in which remodeling/reorganization that mediates recovery of movement in the paretic side is likely to be found.

We replaced the mentioned sentence by the following phrase: "As we used a large sensorimotor stroke our study focused on the manipulation and circuit investigation of the intact, contralesional hemisphere as the main region where plastic remodeling and reorganizational processes are likely to be found which mediate recovery of movements in the paretic side."

Minor

The authors might consider separately reporting the EMG responses for wrist, elbow and shoulder that were evoked from the M1 (Pos. 2) hotspot, for example, in a supplementary table.

We included Supplementary Figure 7 showing heat maps of EMG responses in wrist, elbow and shoulder for the Anti-Nogo/Training animals and animals of the spontaneous recovery group.

The authors might inform the reader that the right side is the paretic side in the Supplementary movie.

We included an explanation of the supplementary movie in Supplementary Materials indicating the stroke side (left) and the right-sided paw paresis.

The blue line that presumably represents the OptoStim group in Fig 2C is dissimilar to its defined color code.

We thank for this hint and adapted the color code of the OptoStim group in Fig. 2C.

There continues to be a need for more careful editing.

- *Caption to Fig. 1, the abbreviation for primary motor cortex is defined as "M2"*
- *Caption to Supp Fig. 1: "coronar" should be "coronal", "secondary" should "secondary"*
- *Second to last paragraph of p. 10, "ilpsilaterally" should be "ipsilaterally" and "innovation" should be "innervation".*
- *New text on p. 16, "subjection" should be "subjective"*

Again many thanks for all these hints. We corrected all misspellings mentioned.

Reviewer #5 (Remarks to the Author):

I am satisfied with the changes the authors made to the manuscript. Figures are clearer and all claims are proportioned to the findings.

I think uploading the software to github is definitely the best option - as long as the github link appears in the accepted manuscript, of course.

We will upload the software to github once the manuscript is accepted.

Also, I strongly recommend embedding the control shown in Supplementary Figure 4 directly in Figure 3. There is no reason for it to be relegated to supplementary results.

We tried to embedd Supplementary Figure 4 in Figure 3. But this attempt resulted in a significantly decreased size of all sub-figures and images considerably downgrading readability and visibility. Thus we decided to keep Supplementary Figure 4 and Figure 3 separately.

We thank all reviewers for the helpful comments and hope that we have now addressed all comments and concerns.

REVIEWERS' COMMENTS:

Reviewer #1 (Remarks to the Author):

in this revised manuscript, the authors have addressed all of my concerns.

Reviewer #4 (Remarks to the Author):

The authors have done a very nice job of addressing the minor issues that I raised in the last review and I have no further comments.

Our responses to the referees' comments are as follows:

Reviewer #1 (Remarks to the Author):

in this revised manuscript, the authors have addressed all of my concerns.

Reviewer #4 (Remarks to the Author):

The authors have done a very nice job of addressing the minor issues that I raised in the last review and I have no further comments.

We thank reviewer #1 and #4 for their helpful comments and are glad that our responses satisfied their concerns.